# Germline modifiers of the tumor immune microenvironment implicate drivers of cancer risk and immunotherapy response

Meghana Pagadala [1], Timothy J. Sears [2], Victoria H. Wu[3], Eva Pérez-Guijarro [4], Hyo Kim[5], Andrea Castro[2], James V. Talwar [2], Cristian Gonzalez-Colin[6], Steven Cao[7], Benjamin J. Schmiedel [6], Shervin Goudarzi[8], Divya Kirani [9], Jessica Au [2], Tongwu Zhang [10], Teresa Landi [10], Rany M. Salem [7], Gerald P. Morris [11], Olivier Harismendy[2,12], Sandip Pravin Patel[13], Ludmil B. Alexandrov[14,15], Jill P. Mesirov[16,17], Maurizio Zanetti[16,18], Chi-Ping Day [4], Chun Chieh Fan [19,20], Wesley K. Thompson[21], Glenn Merlino[4], J. Silvio Gutkind[3], Pandurangan Vijayanand [6] & Hannah Carter [16,17] ✉

With the continued promise of immunotherapy for treating cancer, understanding how host genetics contributes to the tumor immune microenvironment (TIME) is essential to tailoring cancer screening and treatment strategies. Here, we study 1084 eQTLs affecting the TIME found through analysis of The Cancer Genome Atlas and literature curation. These TIME eQTLs are enriched in areas of active transcription, and associate with gene expression in specific immune cell subsets, such as macrophages and dendritic cells. Polygenic score models built with TIME eQTLs reproducibly stratify cancer risk, survival and immune checkpoint blockade (ICB) response across independent cohorts. To assess whether an eQTL-informed approach could reveal potential cancer immunotherapy targets, we inhibit *CTSS*, a gene implicated by cancer risk and ICB response-associated polygenic models; *CTSS* inhibition results in slowed tumor growth and extended survival in vivo. These results validate the potential of integrating germline variation and TIME characteristics for uncovering potential targets for immunotherapy.

Cancer is a disease characterized by heterogeneous somatic and germline mutations that promote abnormal cellular growth, evasion from the immune system, dysregulation of cellular energetics, and inflammation[1–4]. Both inflammation and immune surveillance contribute to the selective forces that shape tumor evolution[3–6]. Immunotherapies alleviating immune suppressive signals have emerged as a promising treatment strategy; however, response rates are low and the determinants of response remain elusive[7,8]. Furthermore, the potential of galvanizing the immune system is still unmet due to an incomplete understanding of the complex tumor immune microenvironment

(TIME). In particular, knowledge of germline factors and other intrinsic factors that interact with characteristics of tumors to render them sensitive to host-immunity or immunotherapy is lacking.

Germline variation is responsible for a considerable proportion of variation in immune traits in healthy populations[9,10]. In the context of tumors, germline variants are associated with immune infiltration, antigen presentation and immunotherapy responses[11,12]. Autoimmune germline variants modify immune checkpoint blockade (ICB) response and variants underlying leukocyte genes predict tumor recurrence in breast cancer patients[13,14]. For example, the common single nucleotide

polymorphism (SNP) rs351855 in *FGFR4* was found to suppress cytotoxic CD8+ T cell infiltration and promote higher immunosuppressive regulatory T cell levels via increased *STAT3* signaling in murine models of breast and lung cancer[15]. Normal genetic variation underlying major histocompatibility complex molecules, MHC-I and MHC-II, dictate which mutations in an individual's tumor can elicit immune responses, and play a role in antigen-driven host anti-tumor immune activity that influences tumor genome evolution through immune selection[16,17]. Polymorphic variation in these regions has also been linked to treatment outcomes[18–20]. Recent literature highlights polymorphisms in other immune-related genes such as *CTLA-4*[21], *IRF5*[22], and *CCR5*[23,24] that also affect treatment outcomes.

Efforts to identify germline variation associated with anti-tumor immune responses have pointed to effects on immune infiltration levels and immune pathways, such as *TGF-β* and *IFN-γ*[11,12,25]. Genes with significant cis-eQTLs in the TCGA are both enriched for immune-related genes and associated with immune cell abundance within the TIME[26]. These studies provide evidence that variants may act through specific effects on immune cells. eQTL profiling of 15 sorted immune cell subsets from healthy individuals found that the effects of many eQTLs were specific to immune cell subsets[27]. Understanding mechanisms and cell-type effects of TIME host genetic interactions could not only identify aspects of immunity that negatively impact cancer and immunotherapy outcomes, but also point to putative targetable cell types and molecules for modulating immune responses.

In this work, we identify common germline variants associated with TIME characteristics that are also associated with cancer outcomes, reasoning that such dual associations would implicate the aspects of immunity most critical for tumor control and uncover putative targets for immunotherapy[28,29]. We construct and validate polygenic models to predict cancer risk, survival and ICB response, studying the eQTLs selected during model fitting to gain functional insights. Our results support a role for common immune variants in cancer risk, survival and immunotherapy response, and provide a potential strategy for immunotherapy target discovery. The study design is summarized in Fig. 1A.

## Results

### Identifying heritable characteristics of the tumor immune microenvironment

To focus on common germline genetics with the potential to modify tumor immune responses, we assessed which characteristics of the tumor immune microenvironment (TIME) showed evidence of SNP heritability. To describe the TIME, we collected a comprehensive set of immune phenotype ("IP") components comprising composite measures derived from bulk gene expression and expression levels of individual immune-related genes (Fig. 1B). Composite phenotypes included infiltrating immune cell levels calculated using CIBERSORTx (immune infiltrates) and 6 immune subtype scores from a pan-cancer TCGA analysis by Thorsson et al. (landscape components). Immunomodulators were collected from Thorsson et al., where weighted gene correlation network analysis was used as an unbiased systematic approach to identify gene sets relevant to the TIME. We included genes from these sets along with immune checkpoint genes, cell type markers, antigen presentation genes, TGF-β pathway genes, and IFN-γ genes as these have been implicated as important modifiers of the TIME. After removing IP components with high numbers of zero values to reduce spurious associations, we retained 724 immune-related genes and 9 composite phenotypes (733 IP components total) measured across 30 cancer types (Supplementary Data 1–3 and Supplementary Fig. 1). Each IP component (gene expression level or composite phenotype) was analyzed independently.

We evaluated the potential of germline variation to explain inter-tumor differences in IP components by performing SNP heritability analysis (Fig. 1A). Since highly polymorphic regions such as the HLA locus can inflate SNP heritability estimates[30], we separately estimated SNP heritability attributable to the HLA locus and the rest of the genome. We identified 235 (32.0%) IP components where levels were SNP-heritable (Fig. 1C and Supplementary Data 4). No composite phenotypes passed heritability thresholds and thus the remaining associations were with gene expression and will be referred to as TIME eQTLs. For these 235 genes, we conducted 2-state GCTA analysis and identified 140 (59.6%) that had a significant proportion of SNP heritability attributable to regions outside the HLA locus, while 17 (7.2%) were mostly attributable to the HLA locus. We focused our TIME eQTL discovery analysis on these 157 heritable immune genes.

To assess the possibility of tumor-type specific SNP-heritable effects, we revisited the SNP-heritability analysis in breast cancer, which had the most samples. The 2-state heritability analysis uncovered 17 genes (FDR < 0.05), including HLA region genes (*HLA-A*, *HLA-C*, *HLA-G*, *HLA-DRB1*, *HLA-DRB5*, *HLA-DQB1*, *HLA-DQB2*, *MR1*, *MICA*, *BTN3A2*, *HLA-DQA2*, *HLA-DQA1*, and *PAICS*) and *ERAP2* and *DCTN5 genes which* were shared with the pancancer analysis. Two additional genes, *KRR1* and *FN1*, were only detected in the breast cancer-specific analysis. *FN1* encodes fibronectin, which plays a role in the stromal microenvironment and tumor invasion[31]. It has been implicated in development of several tumors, including breast cancer[32,33]. *KRR1* is a proteasomal subunit linked to integrin expression in breast cancer[34]. These results suggest that there are likely shared heritable features related to antigen presentation, but also differences that could be unique to each cancer's microenvironment (Supplementary Data 5). However, larger sample sizes are needed to investigate tumor-type specific effects.

### Detecting putative germline modifiers of the TIME

To identify TIME eQTLs, we performed a genome-wide association study (GWAS). First, we analyzed each of the 140 heritable immune genes outside of the HLA locus across individuals of European ancestry in the TCGA (Supplementary Fig. 2A). Immune gene expression was inverse-rank normalized within tumor type, such that tumor-type specific differences were removed (Supplementary Fig. 1). Only common germline variants with minor allele frequency >1% were considered and imputation quality (Rsq) was evaluated to ensure high accuracy (Supplementary Fig. 2B). No evidence of inflation was observed (Supplementary Fig. 2C). Using linkage and distance-based clumping[35], we identified 825 TIME eQTLs (Fig. 2A, Supplementary Data 6). *Cis* associations, defined as an associated locus occurring within 1 MB of a gene transcription start site, encompassed the majority (95.0%) of associations[36], while 5.0% of the associations were *trans*. Mechanisms of *trans* associations are complex and tend to have weaker effects on transcriptional regulation[37]. In contrast, *cis* associations are proximal to an IP component and have more direct effects on transcription. Overall, *ERAP2* (181, 21.9%), *CCBL2* (76, 9.2%), *DHFR* (75, 9.0%), and *ERAP1* (70, 8.5%) had the most germline associations (Supplementary Fig. 2D) of the 140 genes tested.

To remove HLA region associations solely attributable to LD structure[38,39], we conducted conditional GWAS analysis for seventeen genes in the HLA region of chromosome 6. Alignment to a general HLA gene reference can introduce error into expression level estimates due to the highly polymorphic nature of these genes. We therefore also revisited SNP associations with gene expression estimates derived from allele-specific RNA alignments[40] ("Methods" section) and performed GWAS analysis using allele-specific expression. In total, we identified 65 TIME eQTLs in the HLA region (Fig. 2B). Combining GWAS and conditional HLA GWAS associations, we identified 890 TIME eQTLs. Generally, LD-independent eQTLs clustered by genomic regions with *HLA-A*, *HLA-B*, *HLA-C* associated variants falling in the MHC Class I genomic region and *HLA-DQB1*, *HLA-DQA1*, *HLA-DPB1*, *HLA-DRB5* associated variants falling in the MHC Class II genomic region (Supplementary Fig. 2E). We note that *HLA-DRB5* only occurs on

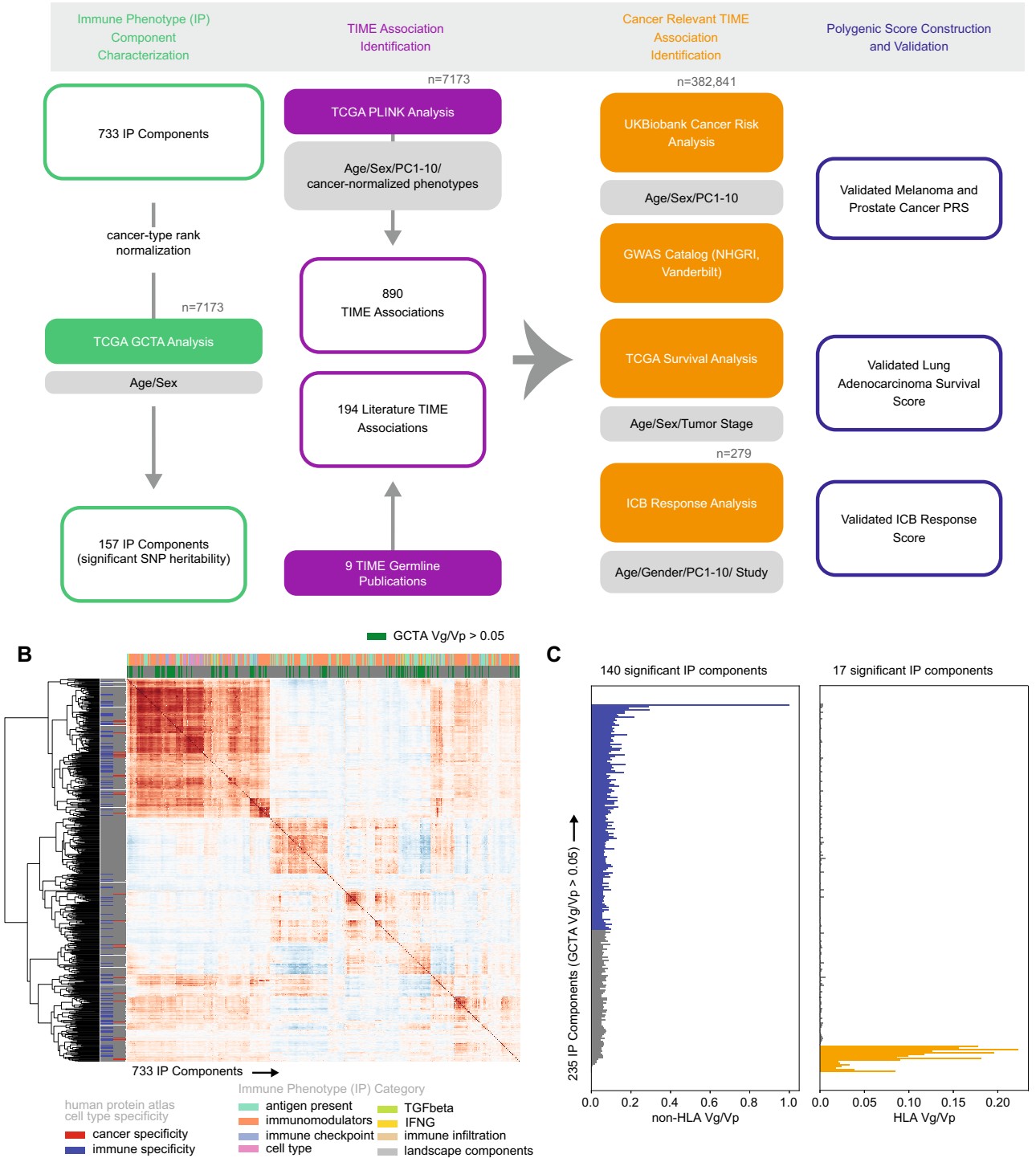

**Fig. 1 | Identifying heritable characteristics of the tumor immune microenvironment. A** Overview of the TIME germline analysis. Sample sizes are shown in gray. **B** Clustermap depicting 733 IP components and their pairwise correlation across 30 tumor types in the TCGA[174]. **C** Horizontal barplot of variance in phenotype explained by variance in genotype (Vg/Vp) for 235 immune genes estimated separately genome-wide excluding the HLA locus (left panel, blue) and using only the HLA locus (right panel, orange). Source data are provided in the Source Data file.

specific haplotypes, but has homology to *HLA-DRB3* and *HLA-DRB4* which could lead to erroneous assignment of gene expression in individuals where the *HLA-DRB5* gene is absent. We therefore revisited eQTL analysis for HLA-DRB5 using only individuals with *HLA-DRB1*15* and *HLA-DRB1*16* alleles, which indicate haplotypes inclusive of the *HLA-DRB5* gene[41–43]. This analysis implicated 2 SNPs associated with *HLA-DRB5* expression levels. (Supplementary Data 7).

We noted some correlation among immune genes across tumors, especially those related to macrophages and lymphocytes which were the most abundant infiltrating immune cells (Supplementary Fig. 2F). The largest group of correlated genes included MHC Class I and II genes along with macrophage genes *VSIG4, CD163, FCGR2A FCGR3A, HAVCR2, LILRB2, LILRB4*, and *CD53* (Supplementary Fig. 3A and S3B) and was most strongly associated with antigen presentation, dendritic cell processing, and *IL-10* production (Supplementary Fig. 3C). The next largest comprised two anti-correlated gene subgroups which contained *EP300* and *TREX1* respectively (Supplementary Fig. 3D) and was related to innate

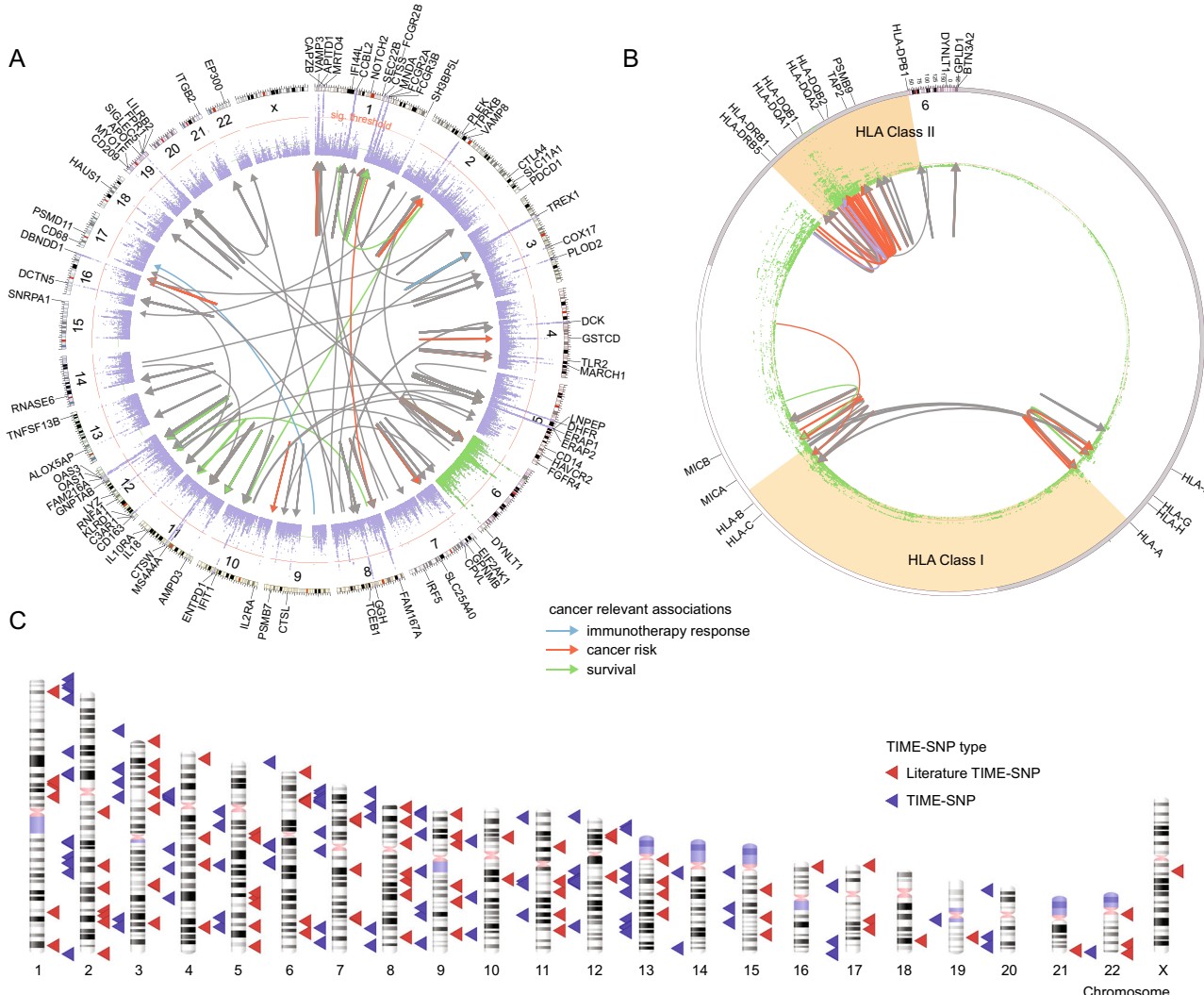

**Fig. 2 | Detecting putative germline modifiers of the tumor immune microenvironment. A** Locuszoom plot summarizing 890 associations between TIME eQTLs and 93 immune genes. Outer ring represents locations of all 157 tested IP components. Links are colored if implicated in cancer risk (orange), survival (green), or immunotherapy response (blue). **B** Significant associations between TIME eQTLs and 17 genes in the HLA region detected through conditional GWAS analysis for effects on gene expression using either a basic alignment to the reference genome (conditional), or allele-specific expression obtained by aligning to a patient-specific HLA reference allele set. **C** Ideogram plot of TIME eQTLs implicated by our discovery analysis (red) and literature curation (blue). Source data are provided in the Source Data file.

immune activation, the C-type lectin receptor signaling pathway and antigen presentation (Supplementary Fig. 3E). These two groups correlated strongly with the top 2 principal components from Principal Component Analysis (PCA) conducted on the expression of the 157 unique SNP-heritable immune genes. *CD53, CD86*, and *CYBB*, which are highly correlated (ρ > 0.7) to the Thorsson et al.[44] macrophage regulation score, were major contributors to PC1 while *HACD2, LNPEP*, and *EP300* were major contributors to PC2 (Supplementary Data 8). We investigated whether this gene correlation would inflate the chance of detecting eQTLs associated with a particular group, however analysis of summary statistics showed that despite their correlation, genes typically did not recover the same SNP associations unless they were encoded at the same genomic locus, such as *ERAP1* and *LNPEP* or *OAS1* and *OAS3* (Supplementary Fig. 3F). Finally, to confirm TIME eQTLs were not cancer-type specific, we conducted associations with tumor type. Of our 890 TIME eQTLs, only rs146336885 was associated with tumor type (Supplementary Data 9 and Supplementary Fig. 3G).

Previous studies of germline variation and important modulators of immune checkpoint response such as *APOE*[45], *CTSW*[46], *CTLA-4*[21],

*PD-L1*[47,48], *PD-1*[49–51], *CXCR3/CCR5*[23], *IRF5*[22], and *FGFR4*[15] along with immune signatures and immune cell infiltration have been conducted[11,26,52]. We incorporated these 194 germline associations from literature into our analyses (Fig. 2C and Supplementary Data 10). Like Shahamatdar et al.[11], we included immune infiltrates estimated from bulk RNA sequencing into the set of immune components we investigated, however, none of the CIBERSORTx infiltrates passed our SNP-heritability filter. Zhang et al.[46]. took a fundamentally different approach, analyzing ER + breast cancer-associated variants from Michailidou et al.[53] for proximity to immunoinflammatory GWAS variants. The top SNP, rs3903072, was an eQTL for *CTSW* in breast cancer. Although not specifically focused on breast cancer, our study also identified *CTSW* as a SNP-heritable IP component (GCTA V(g)/ V(p) = 12.1%) and detected a pan-cancer association with rs3903072 (beta=0.21, *p* = 2.8e-36). The study by Sayaman et al.[52] focused on 139 immune traits described in the Thorsson et al.[44] paper, of which 106 were immune signatures and 33 included immune measures such as TCR/BCR characteristics, CIBERSORTx infiltration and antigen load. Comparing gene results between Sayaman et al. and our study, 10 genes were shared between our analyses, *HLA-DRB5, HLA-B, HLA-DRB1,*

*MICB, HLA-DQB1, HLA-DQB2, HLA-DQA1, HLA-DQA2, MICA, HLA-C*, emphasizing the importance of MHC Class I and II machinery in modifying the TIME. Nineteen of our variants were in LD with 361 Sayaman et al. TIME eQTLs ($R^2 > 0.50$).

Combining our TIME eQTLs and literature associations resulted in a set of 1084 candidate associations. A number of TIME eQTLs were associated with multiple immune genes; thus, we had a greater number of associations than TIME eQTLs. For example, within our own discovery pipeline rs2693076 was associated with *LILRB2, PLEK, MYO1F*, and *CD14*. From literature curation, Sayaman et al. identified associations with rs2111485 and multiple signatures, including interferon-signaling and *IFIT3* signaling.

### Identification of TIME eQTLs related to cancer outcomes

We next wanted to determine if TIME eQTLs could serve as the basis for genetic models for cancer risk, survival and immunotherapy response prediction. An association with gene expression in the TIME does not necessarily mean that the eQTL will impact cancer outcomes. Thus, we evaluated our TIME eQTLs in the context of human cohorts, relying on datasets with both genetic and relevant cancer phenotype data to build models.

For cancer risk, we performed a PheWAS with cancer ICD10 codes in the UK Biobank, and also cross-referenced our associations against summary statistics from the NHGRI-EBI GWAS catalog[54,55] and Vanderbilt PheWAS catalog[56] (Supplementary Data 11). We observed high overlap in risk variants (FDR < 0.05) identified by these three sources (Supplementary Fig. 4A). When assessing overlap based on the corresponding genes, an even higher degree of overlap was observed, with only 2 eQTLs, *TAP2*, and *LNPEP*, being uniquely implicated by the UK Biobank (Supplementary Fig. 4B). For survival analysis, we evaluated TIME eQTLs with overall and progression-free survival in the TCGA dataset, treating each tumor type separately. Survival association was evaluated by CoxPH model for tumor types with at least 100 samples available and including covariates relevant to each tumor type (Supplementary Data 12–13).

To investigate the implication of TIME eQTLs for immune checkpoint blockade (ICB) response, we collected sequencing and ICB response information for 279 patients with melanoma treated with immune checkpoint inhibitors from 4 studies[57–61], and imputed SNPs from exome sequencing data. PCA analysis of genotypes showed no batch effects (Supplementary Fig. 4C). Accuracy of exome-based imputation was assessed by comparing original TCGA genotype calls to genotypes imputed in from TCGA exome data at positions matching those in the ICB data; aside from variants on chromosome 6 within the HLA region most were accurately imputed (Supplementary Data 14). Ultimately, 525 out of 1084 TIME eQTLs could be imputed with sufficient quality (minor allele frequency >0.05 in all 4 discovery ICB cohorts with imputation accuracy of at least 0.3[62,63] (Supplementary Fig. 4D). We conducted meta-analysis with METAL[64] using the four melanoma ICB cohorts to evaluate SNP associations with ICB response. No individual eQTLs were significantly associated with ICB response after multiple testing correction (Supplementary Data 15).

To model the role of immune genetic background in cancer phenotypes as a whole, we used polygenic scores. We adopted the polygenic score construction approach by Elgart et al.[65] which performs shrinkage-based SNP selection followed by construction of a nonlinear, machine learning based PRS capable of capturing interactions between SNPs. For risk analysis, we selected two cancer types for more in depth analysis. We repeated our survival analysis with tumor-type specific polygenic survival score (PSS) as the independent variable. We also constructed a polygenic ICB score (PICS) in the four ICB melanoma cohorts. In each case, we validated genetic models in independent cohorts. These analyses are described below.

### TIME eQTLs underlying antigen presentation stratify melanoma and prostate cancer risk

To assess the potential of immune genetic background to influence cancer risk, we evaluated TIME eQTL derived polygenic risk scores (PRSs) in two cancer types with differing levels of immune involvement. Melanoma is classically thought of as an immune 'hot' cancer type, with high levels of immune infiltration and one of the highest rates of immunotherapy response[66]. In contrast, prostate cancer tends to have a more suppressed immune microenvironment[67,68].

We first constructed PRS from TIME eQTLs in UK Biobank separately for melanoma and prostate cancer. Because TIME eQTL risk associations were derived in part from the UK Biobank, we sought to evaluate the resulting PRS models in independent cohorts. We validated the melanoma PRS in 3029 melanoma cases and controls from UT MD Anderson[69]. As is typical for PRS scores, the difference in score distributions for cases and controls was small (Fig. 3A), but the odds of melanoma were significantly different in the top and bottom 10th quantile in the validation cohort (Fig. 3B). eQTLs related to *CTSS* and MHC class II genes featured prominently among the most informative features during model fitting, suggesting a role for class II antigen presentation in cancer risk (Fig. 3C). We validated the prostate cancer PRS in a cohort comprising 91,644 cases and controls from the ELLIPSE Consortium[70] with similar results (Fig. 3D, E). CTSS and class II MHC genes were once again the most important features, though *HLA-B* and *HLA-C* appeared more influential in prostate cancer risk (Fig. 3F). Effect sizes separating the top and bottom quantiles were larger in melanoma than prostate cancer (Fig. 3B vs E and Supplementary Fig. 5). While pan-cancer risk analysis implicated individual eQTLs for *CTSS, ERAP1, ERAP2, CTSW* and class I and II MHC genes (Supplementary Data 11), PRS analysis pointed to additional eQTLs with some shared between melanoma and prostate (*FPR1, LYZ, FCGR3B, HLA-G, HLA-H, HLA-DQA1*, and *HLADQB1*), unique to melanoma (*MNDA, IL2RA, OAS1*, and *TAP2*) or unique to prostate (*AMP3D, SIGLEC5, HLA-B, HLA-C*, and *HLA-DRB1*).

As the PRS analysis implicated aspects of both antigen directed T cell responses and macrophage activity, we asked whether the melanoma PRS correlated with T cell and macrophage phenotypes in melanomas in the TCGA dataset. Indeed, tumors in the upper 10th quantile of the melanoma PRS had higher levels of infiltration by pro-tumor inflammatory M2-like (Fig. 3G), but not M0 or M1-like macrophages. Promotion of an inflammatory pro-tumor environment was also correlated with decreased CD8+ T cell infiltration (Fig. 3H). This supports that TIME eQTLs contribute to cancer risk at least in part by modifying the activity of immune cells at the site where a tumor develops.

### TIME eQTLs associated with survival implicate immune evasion

We also revisited survival associations to evaluate polygenic contributions. We built cancer type-specifc PSS separately for each tumor type using 70% of samples, then used them to calculate PSS for the remaining 30% of tumors, and evaluated these scores along with other covariates in a Cox Proportional Hazards analysis. We found significant associations with overall survival in lung adenocarcinoma, stomach adenocarcinoma, bladder urothelial carcinoma, breast invasive carcinoma, clear cell renal carcinoma, papillary renal carcinoma, head and neck squamous cell carcinoma, lung squamous cell carcinoma, esophageal carcinoma, pancreatic adenocarcinoma, rectal carcinoma, colorectal adenocarcinoma (FDR < 0.05; Fig. 4A) and with progression-free survival in lung adenocarcinoma, breast invasive carcinoma, bladder urothelial carcinoma, rectum adenocarcinoma, colorectal adenocarcinoma, pancreatic adenocarcinoma, stomach adenocarcinoma, and hepatocellular carcinoma (FDR < 0.05; Fig. 4B).

Among these tumor types, we were able to obtain matched survival and genotype data for non-smokers that developed lung cancer from the Sherlock cohort[71]. PSS-stratification of the 30% of TCGA lung adenocarcinoma (LUAD) samples (Fig. 4C) and individuals in the Sherlock cohort (Fig. 4D) showed similar effects on outcome, such that

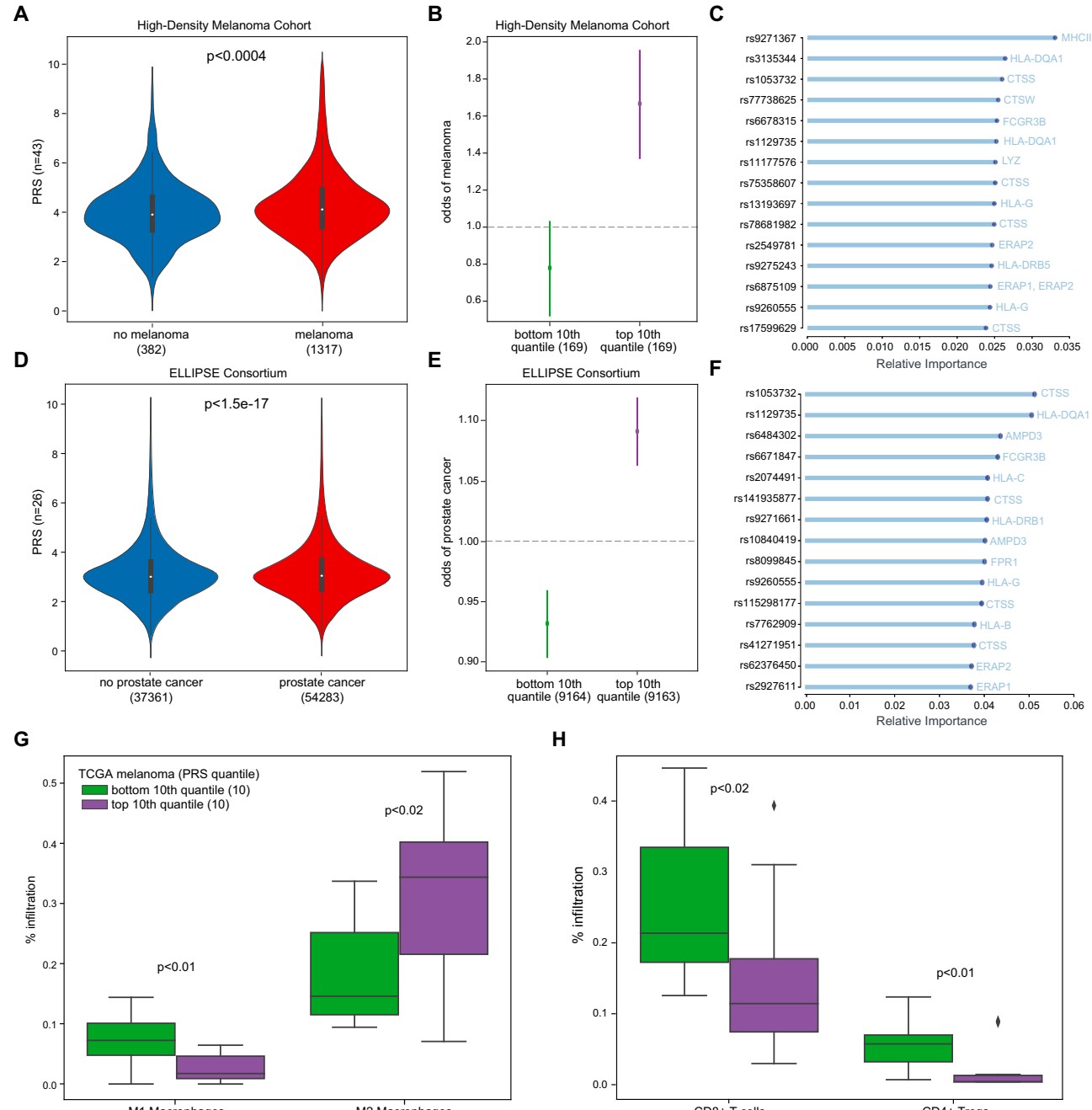

**Fig. 3 | TIME eQTLs underlying antigen presentation stratify melanoma and prostate cancer risk. A** Violinplot of melanoma PRS trained on UK Biobank[175] and validated in 1317 melanoma cases and 382 controls in High Density Melanoma cohort[69,157]. **B** Logistic regression odds ratio of melanoma risk ±SE among individuals in the top and bottom 10th quantile of PRS in High Density Melanoma cohort. **C** Top 15 TIME eQTL features most important in melanoma PRS. **D** Violinplot of prostate cancer PRS trained on UK Biobank and validated in 54,283 prostate cancer cases and 37,361 controls in ELLIPSE Consortium. **E** Logistic regression odds ratio of prostate cancer risk ±SE among individuals in the top and bottom 10th quantile of

PRS in ELLIPSE consortium[158]. **F** Top 15 TIME eQTL features most important in prostate cancer PRS. **G** Boxplot of M1 and M2 macrophage infiltration in primary TCGA SKCM (melanoma) in top ($n = 10$) and bottom ($n = 10$) 10th quantile of melanoma PRS. **H** Boxplot of CD8+ T cell and CD4+ T regulatory cell infiltration in TCGA SKCM (melanoma) in top and bottom 10th quantile of melanoma PRS. Boxplots show median (line), 25th and 75th percentiles (box), and 1.5× the interquartile range (IQR, whiskers). Two-sided Mann–Whitney $U$ p-values were used for comparisons, adjusted if >2 comparisons were being made.

tumors with the lowest PSS scores had the best overall survival. Incorporating the TCGA LUAD-based PSS into a CoxPH analysis of the Sherlock tumors including clinical covariates (Supplementary Data 13) returned a larger hazard ratio than in the held out 30% of TCGA samples (Fig. 4E). The PSS for overall survival included eQTLs for genes involved in regulating T cell activity (*CTSW, PD-1, PD-L1*), antigen processing and presentation (*VAMP3*[72], *ERAP2, MICA*), response to

immunogenic stimuli such as aberrant DNA or microorganisms (*TREX1, OAS1, C3AR1, FPR1*), suppression of myeloid cells (*SIGLEC5*), folate metabolism (*GGH, DHFR*), amino acid metabolism (*CCBL2*), and interferon signatures (Fig. 4F). The presence of *GGH* and *DHFR* suggested the possibility that our eQTL set could include pharmacogenomic modifiers of anti-folate treatments such as methotrexate and pemetrexed. We therefore revisited our validation analysis, omitting eQTLs

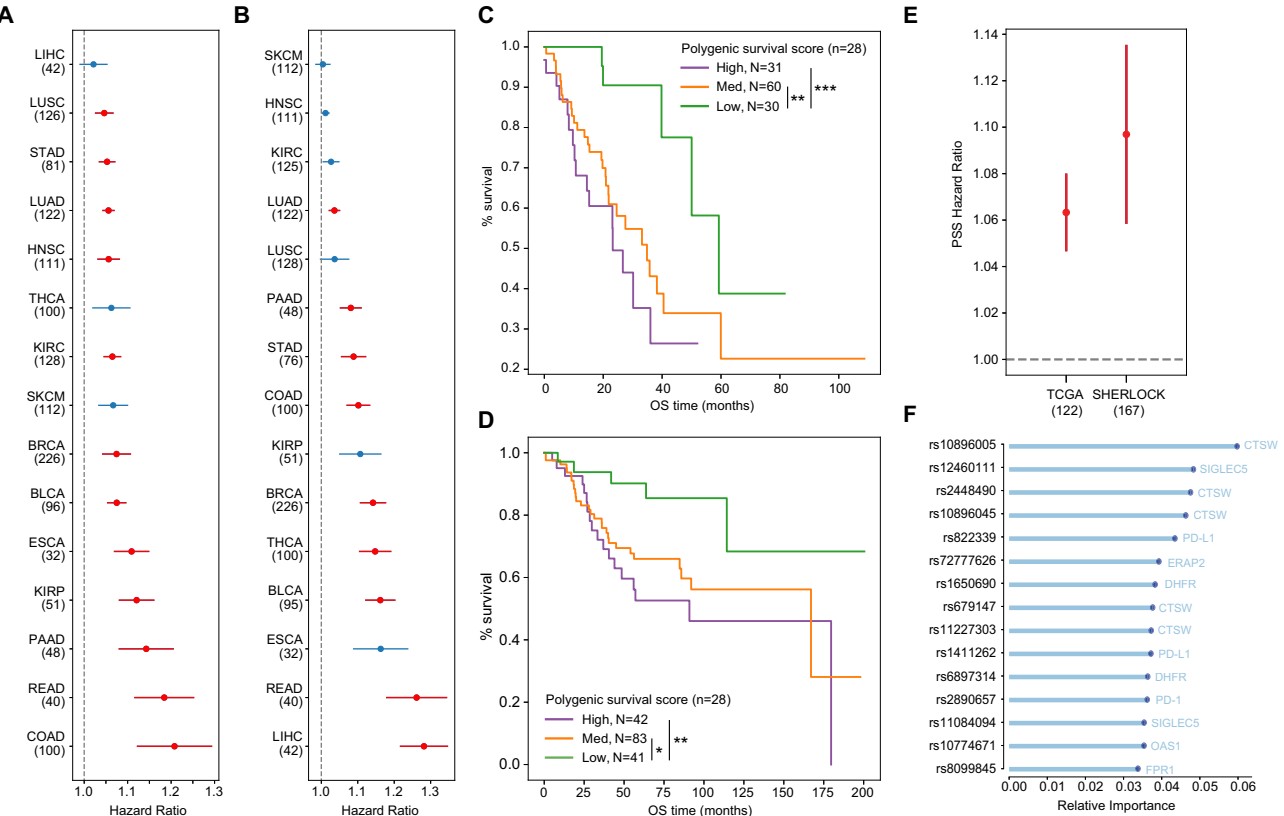

**Fig. 4 | TIME eQTLs associated with survival implicate immune evasion. A** Cox Proportional Hazards ratios ±SE for cancer type-specific polygenic survival score (PSS) with overall survival separated by TCGA[174] cancer type. Sample sizes used for PSS evaluation are indicated in parentheses. Red color indicates that the PSS hazard ratio was significant with an FDR < 0.05 after correcting for the number of tumor types modeled. **B** Cox Proportional Hazards odds ratios ±SE for cancer type-specific PSS with progression-free survival separated by TCGA cancer type. **C** Overall survival Kaplan–Meier curve based on LUAD PSS in TCGA LUAD (n = 121).

**D** Overall survival Kaplan–Meier curve based on LUAD PSS in SHERLOCK[71] (n = 166). **E** Cox Proportional Hazards ratio ±SE for LUAD PSS in TCGA LUAD and SHERLOCK. **F** Top 15 TIME eQTL features most important in the LUAD PSS. For **C** and **D**, High indicates top 25%, Med indicates middle 50% and Low indicates lowest 25% of PSS. Error bars represent standard error of Cox Proportional Hazards model. Significance is marked as: *p < 0.05, **p < 0.01, ***p < 0.001. Source data are provided in the Source Data file.

for these genes, and found that the PSS still validated in Sherlock (Supplementary Fig. 6).

## TIME eQTLs implicate targets for modulating immune responses

We next constructed an immunotherapy response-specific PRS using four published melanoma cohorts treated with immune checkpoint blockade. We validated the predictive potential for this polygenic score in two independent cohorts, one consisting of renal cell carcinomas, and the other of non-small cell lung cancers. In both cohorts, responders had significantly higher polygenic ICB scores (PICS) (Fig. 5A, B) and in ROC analysis the PICS achieved an area under the curve >0.7 (Fig. 5C). Feature importance analysis of the PICS model suggested eQTLs involving genes related to DNA replication (*TREX1, DHFR)* and antigen presentation (*PSMD11, ERAP1, ERAP2, CTSS*) were most informative (Fig. 5D).

Although tumor-immune interactions vary across tissue sites and tumor characteristics, our study design emphasized tumor-general effects which may explain the generalization of the PICS across ICB cohorts with distinct tumor types. The PICS selected 30 TIME eQTLs (Fig. 5E) and one SNP associated with $T_{fh}$ infiltration levels. The PICS implicated genes associated with antigen processing and presentation (*CTSS, ERAP1, ERAP2, PSMD11*), complement (*C3AR1*) and cytolytic activity (*CTSW*), vesicular transport (*DCTN5, DYNLT1*), post-translational regulation (*DBNND1, GPLD1*), folate metabolism (*DHFR*), phagocytic activity (*FPR1, LYZ*), and single-stranded DNA response

(*TREX1)*. We repeated this analysis selecting 31 TIME eQTLs at random, matched for minor allele frequency, and found that the observed difference in burden score between responders and nonresponders was significantly larger than random in both discovery and validation sets (Supplementary Fig. 7A, B). PICS outperformed clinical variables such as age and sex (Supplementary Fig. 7C, D).

For most ICB response genes the direction of effect of variants associated with responder status was mostly consistent across cohorts, though some variants, such as rs28459155 associated with *PSMD11* showed less agreement (Fig. 5E and Supplementary Data 16). rs28459155 associated with lower odds of being a responder in Miao et al., Rizvi et al., and Hugo et al. but higher odds of being a responder in Van Allen et al., Snyder et al., and Riaz et al. As a comparison to current ICB biomarkers, we also evaluated association of tumor mutation burden (TMB) and expression levels of *PD-L1, PD-1,* and *CTLA-4* with responder status and found no significant associations (Fig. 5E). We ran associations with the 31 variants and TMB, *PD-L1, PD-1,* and *CTLA-4* to determine if any variants were associated with these previously researched biomarkers. We observed only an association between TMB and *ERAP1* variant rs27765 and *PD-L1* and *DHFR* variant rs503367 (Supplementary Fig. 7E). Models that used PICS together with TMB and immune checkpoint gene expression had significantly higher variance explained compared to TMB and immune checkpoint gene expression alone (anova p < 0.007; Supplementary Fig. 7F).

We next evaluated PICS-implicated genes as possible entry points to modify anti-tumor immunity. Colocalization of gene expression and

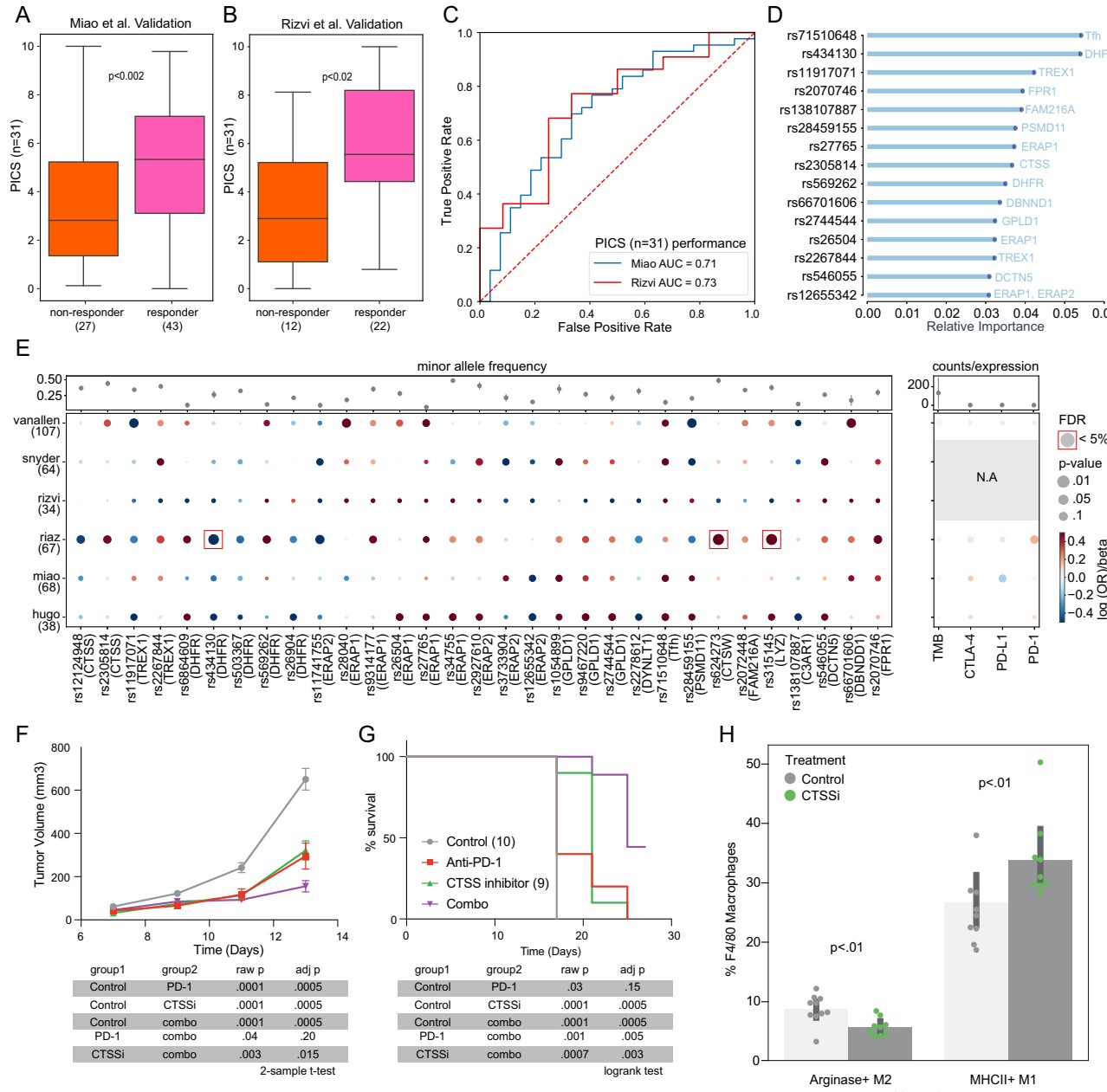

**Fig. 5 | TIME eQTLs implicate targets for modulating immune responses.**
**A** Boxplot of polygenic ICB score (PICS) constructed in melanoma ICB cohort[57,58,60,61] validated in Miao et al. cohort[161] ($n = 70$). **B** Boxplot of PICS constructed in melanoma ICB cohort validated in Rizvi et al. cohort[59] ($n = 34$). **C** ROC-AUC analysis of PICS in Miao et al. and Rizvi et al. validation cohorts. **D** Top 15 TIME eQTL features most important in PICS. **E** Grid plot of log odds ratio of variants with responder status in 6 ICB cohorts with beta coefficients of classic ICB biomarkers (TMB, *PD-L1, PD-1, CTLA-4*) association with responder status. Data are presented as mean values ±SE. Sample sizes for each cohort are indicated in parentheses. **F** Tumor growth curve for C57BL/6 mice implanted with MC38 treated with anti-PD-1, anti-CTSS, and combination of anti-PD-1 and anti-CTSS. Data are presented as mean values ±SE; $n = 10$ mice per group. **G** Survival curve for C57BL/6 mice

implanted with MC38 treated with anti-PD-1, anti-CTSS, and combination of anti-PD-1 and anti-CTSS. **H** Barplot of the proportion of F4/80 Macrophages that are Arginase[+] M2 macrophages and MHCII[+] M1 macrophages respectively for MC38 tumors treated with anti-CTSS compared to control. Data are presented as mean values ±SE. Boxplots show median (line), 25th and 75th percentiles (box), and 1.5× the interquartile range (IQR, whiskers). Two-sided Mann–Whitney *U* p-values were used for PICS comparisons, adjusted if >2 comparisons were being made. Two-sample *t*-test was used for MC38 tumor growth comparisons, Bonferroni-adjusted for multiple tests. Logrank test was used for MC38 survival comparisons, Bonferroni-adjusted for multiple tests. Source data are provided in the Source Data file.

GWAS signals can point to putative causal disease-related genes that in the setting of ICB response might suggest candidate targets to stimulate more effective anti-tumor immunity. Examining gene expression data available for 4 out of 6 cohorts, we noted that none of the 15 genes were significantly differentially expressed between ICB responders and nonresponders (Supplementary Data 17). However, some TIME eQTLs were associated with both higher expression of the

associated gene and worse ICB response, suggesting that these genes could potentially be inhibited to improve anti-tumor immunity (Supplementary Data 18). Of the genes meeting these criteria, only CTSS, TREX1, and PSMD11 had small molecule inhibitors available. For all three genes the effect of the minor allele on gene expression varied across human cohorts (Supplementary Fig. 7G, I). In the Van Allen cohort where rs11917071 associated with lower odds of being a

responder, individuals carrying the minor allele also tended to have increased *TREX1* expression (Supplementary Fig. 7G). In the Hugo et al. and Miao et al. cohorts, individuals carrying rs2267844 trended toward lower *TREX1* expression and higher odds of being a responder (Supplementary Fig. 7H). This is consistent with *TREX1*'s role as an immune inhibitor that prevents cGAS-STRING initiation, with inhibition of *TREX1* stimulating IFN signaling and autoimmunity, making it a potential immunomodulatory target[73,74]. Individuals with rs28459155 had lower odds of being a responder in 3 of the 6 cohorts and trended toward increased expression of *PSMD11* in 2 of these cohorts (Supplementary Fig. 7I). A proteosomal protein involved in ubiquitination, *PSMD11* is associated with worse prognosis in pancreatic cancer[75]. Individuals with *CTSS* variant rs23058814 also had higher odds of being a responder and trended toward decreased *CTSS* expression (Supplementary Fig. 7J). Increased *CTSS* expression has been linked to tumor progression in follicular lymphoma due to decreased CD8+ T cell recruitment[76]. CTSS featured prominently in our cancer risk analysis and, unlike TREX1[73,77], had not been implicated as a likely target for solid tumor immunotherapy. Furthermore, we observed increased M1 macrophage infiltration in individuals with the *CTSS* variant in Hugo et al. suggesting that CTSS activity might contribute to remodeling of the TIME (Supplementary Fig. 7K). These considerations led us to choose CTSS as our top target to validate in vivo. Examining two separate mouse immunotherapy-treated mouse models, we observed significant differences in *Ctss* expression (Supplementary Fig. 7L, M and Supplementary Data 19).

To test the hypothesis that inhibition of CTSS would increase anti-tumor immune activity, we treated mice implanted with MC38 tumors with a CTSS small molecule inhibitor. Mice treated with CTSS inhibitors had slowed tumor growth and better survival compared to control mice (Fig. 5G, H). We also evaluated the interaction of CTSS inhibitor treatment with anti-PD-1. Mice treated with CTSS inhibitor or anti-PD-1 monotherapy had significantly decreased tumor growth and better survival compared to control mice. Additionally, tumor growth was further decreased in mice treated with the combination of anti-PD-1 and CTSS inhibitor as compared to mice treated with anti-PD-1 or CTSS inhibitor alone. In the MC38 model, we observed an increase in infiltrating M1 macrophages and a decrease in M2 macrophages similar to findings from Hugo et al. (Fig. 5I and Supplementary Fig. 8). These findings demonstrate that a focused screen for cancer relevant TIME-associated variants provides a fruitful strategy to reveal novel immunotherapy targets. Furthermore, the influence of CTSS inhibition on the myeloid landscape identifies macrophages as potential cell types that may modulate immunotherapy response.

## Biological implications of TIME eQTLs

Overall polygenic analysis of cancer-relevant TIME eQTLs implicated 91 genes (counting literature-based signatures as a single gene) as potentially contributing to cancer risk, progression or immunotherapy response (Fig. 6A). From these, we sought to understand what aspects of the tumor-immune interface were affected. We evaluated eQTL-implicated genes relative to the two broad functional categories established based on gene ontology enrichment analysis of correlated gene groups in the TIME (Supplementary Fig. 3). While multiple eQTLs in both categories contributed to survival and ICB associations, genes related to innate immune stimulation (Top GO terms: exogenous peptide antigen processing and presentation, NIK/NF-κβ signaling and C-lectin driven innate immune responses) were notably absent from the risk category. This could reflect differences in the tumor types considered in the risk versus survival analyses performed, or it could reflect that such immune eQTLs only become relevant in later stages of disease, perhaps when the right stimuli are present. Literature associations were also mostly tied to progression, possibly reflecting that many of these were originally reported based on observed effects on prognosis.

The majority of TIME-eQTLs were detected as *cis* associations (87.1%), aside from 39 (12.9%) *trans* associations (Supplementary Fig. 9A). Eight cancer relevant TIME eQTLs (1.6%) affected protein-coding regions (Supplementary Fig. 9B). In the case of *HLA−A, HLA-C, FPR1, CTSS* and *TAP2*, missense variants in coding regions were associated with expression differences. In addition, missense variants in PALB2, NOTCH4 and GBP3 were associated with expression differences in *DCTN5*, MHC Class II and *CCBL2*, respectively (Supplementary Fig. 9C).

As the majority of TIME eQTLs fell within non-coding genomic regions, we evaluated their potential to affect regulation of chromatin architecture and transcription based on histone marks[78]. Regions harboring TIME eQTLs were strongly enriched in H3K27ac, H3K36me3 and H3K4me3 and depleted in H3K9me3 (Fig. 6B)[79]. H3K27ac is a known marker of active enhancers and H3K4me3 is usually enriched at promoters near transcription start sites[80,81] suggesting some TIME eQTLs could affect expression of multiple genes while others may be gene specific. TIME eQTLs were depleted in repressive H3K9me3 marks[82]. Enrichment in histone marks was most pronounced in certain immune cell types (Supplementary Data 20).

eQTLs are often cell-type specific[27,83], so we evaluated whether TIME eQTLs in TCGA were dependent on immune cell infiltration level or corresponded to known immune cell-type specific eQTLs in DICE ("Methods" section). Macrophages, CD4+ and CD8+ T cells were the most represented cell types. Of our TIME eQTLs, 48 influenced gene expression in macrophage, 44 were CD4+T cell eQTLs, 42 were CD8+ T cell eQTLs and 27 were B cell eQTLs (Fig. 6C and Supplementary Data 21). Comparing myeloid-specific eQTLs to lymphoid-specific eQTLs, variants associated with *FAM216A, RNASE6, MARCH1, OAS1, HLA-DQB2, GPNMB, LYZ*, and *CPVL* were myeloid-specific.

Re-visiting the 15 genes implicated by the PICS model (Fig. 7), we sought to gain more perspective on the aspects of immunity influential for immunotherapy response. Many of these genes also had risk or survival associated eQTLs and were modifiers of gene expression in various immune cell types. Peptide processing appeared to be a major factor contributing to ICB responses; Peptidases involved in both class I (*ERAP1, ERAP2*) and class II (*CTSS*) peptide processing appeared to be a shared component between ICB response and risk. In contrast, aspects relating to cytolytic activity (*CTSW*), pathogen responses (*FPR1, C3A1*, and *LYZ*) and single stranded DNA responses (*TREX1*) shared more in common between ICB response and progression while eQTLs involving intracellular trafficking proteins DCTN5 and DYNLT1 appeared to uniquely affect ICB response. Interestingly, eQTLs for *DCTN5* showed immune cell type specific effects, whereas those for *DYNLT1* did not. These proteins mediate vesicle and organelle trafficking that may have different implications in different cell types. For example, in T cells they may play a role in immune synapse formation and energetics by transporting mitochondria to the membrane[84]. Interestingly, another vesicle trafficking gene, *VAMP3*, was implicated in progression. Altogether, our analyses reveal a subset of TIME eQTLs that highlight key aspects of immune function with implications for cancer risk, progression, and immunotherapy response.

## Discussion

The success of immunotherapies has generated enthusiasm for using the human immune system as a weapon to eliminate cancers[85–88]. However the very existence of cancer indicates the failure of the immune system to control malignant cell populations throughout multiple stages of tumor development[4]. Here we studied common genetic variants associated with interindividual differences in immune traits and the tumor immune microenvironment, reasoning that these variants could reveal the aspects of immunity most critical for the successful immune control of tumors. Focusing on immune characteristics that showed evidence of SNP heritability in The Cancer Genome Atlas or were implicated in the literature, we screened for

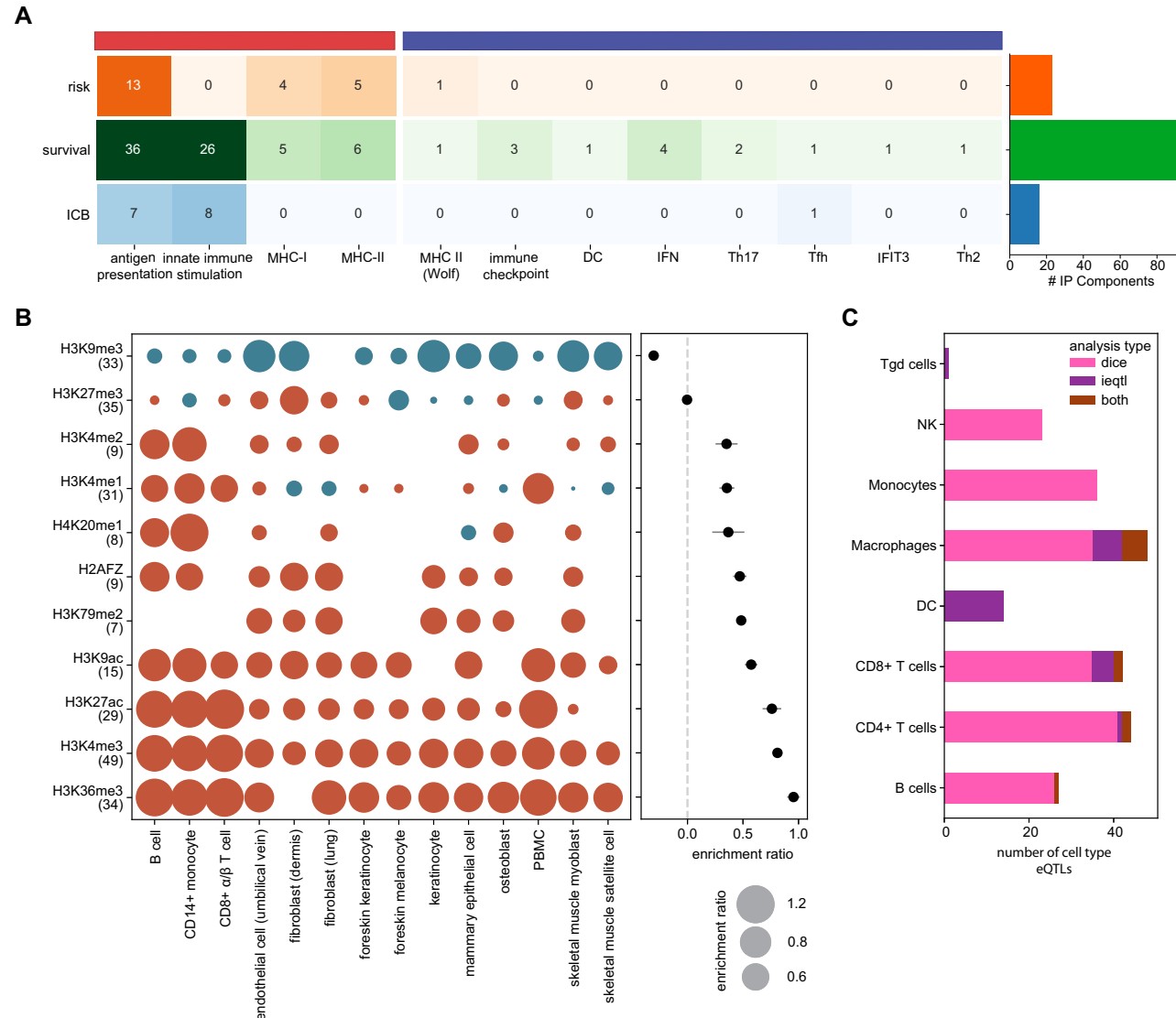

**Fig. 6 | Characterization of TIME eQTLs using in genetic models. A** Cancer relevant associations by category with barplot showing the total number of genes implicated by polygenic risk scores (PRSs), polygenic survival score (PSS) and polygenic ICB score (PICS). **B** Mean enrichment ratio of genetic model immune microenvironment variants in 11 histone marks with corresponding enrichment ratios in 14 specific cell types. Number of cell types analyzed for each histone mark are given. **C** Barplot of cell-type specific TIME eQTLs implicated by DICE and ieQTL analysis. Source data are provided in the Source Data file.

eQTLs in the tumor immune microenvironment. We then used polygenic score analysis to link genes to cancer risk, progression or immunotherapy response via their eQTLs. Although there were many differences in the genes linked to cancer risk, progression and response to immune checkpoint blockade, the 15 associated with ICB response often contributed to predictive models across multiple categories. These included genes related to antigen processing and presentation, innate immunity, and intracellular trafficking.

The immune system interacts with tumors throughout their development and treatment, both through tumor-promoting inflammation and immune-mediated elimination of cancerous cells[89,90]. Adaptive immunity played a significant role across all aspects of our analysis. Alongside multiple MHC I and MHC II genes, TIME eQTLs affected non-HLA antigen presentation pathway genes: *CTSS, CTSW, ERAP1, ERAP2,* and *TAP2. ERAP1* and *ERAP2* are endoplasmic reticulum peptidases that trim peptides before loading them onto MHC proteins[91,92]. *ERAP1/ERAP2* polymorphisms have been associated with cervical cancer and autoimmunity[93–99]. *CTSS* is a cysteine protease critical for MHC Class II loading and is frequently mutated in follicular lymphoma. Its loss limits communication with CD4+ T follicular helper

cells while inducing antigen diversification and activation of CD8+ T cells[76,100]. *CTSW* is crucial for cytotoxicity and is expressed in specific immune cell types[100]. Interestingly, the involvement of MHC II and immune cell specific genes suggest that inter-individual variation in immune surveillance contributes to cancer risk. Notably, we saw that genes implicated in cancer risk were mainly those involved in both MHC Class I and Class II antigen processing and presentation, while TIME eQTLs associated with prognosis pointed to genes that would support evasion of the MHC I CD8+ T cell axis including *PD-L1, PD-1,* and *CTSW.*

Polygenic risk scores for melanoma and prostate cancer, two tumor types falling at opposite ends of the spectrum of immune activity[101,102], both pointed to a role for MHC II-based antigen presentation. MHC Class II expression has been linked to ICB response in melanoma;[103] Although prostate cancer is considered immunologically "cold", rare dramatic responses to immunotherapy have been documented[104]. MHC Class II is usually restricted to professional antigen presenting cells although prostate cancer cells have been shown to express MHC Class II. The Class II pathway is crucial for a prolonged anti-tumor response as it leads to sustained CD8+ T cell

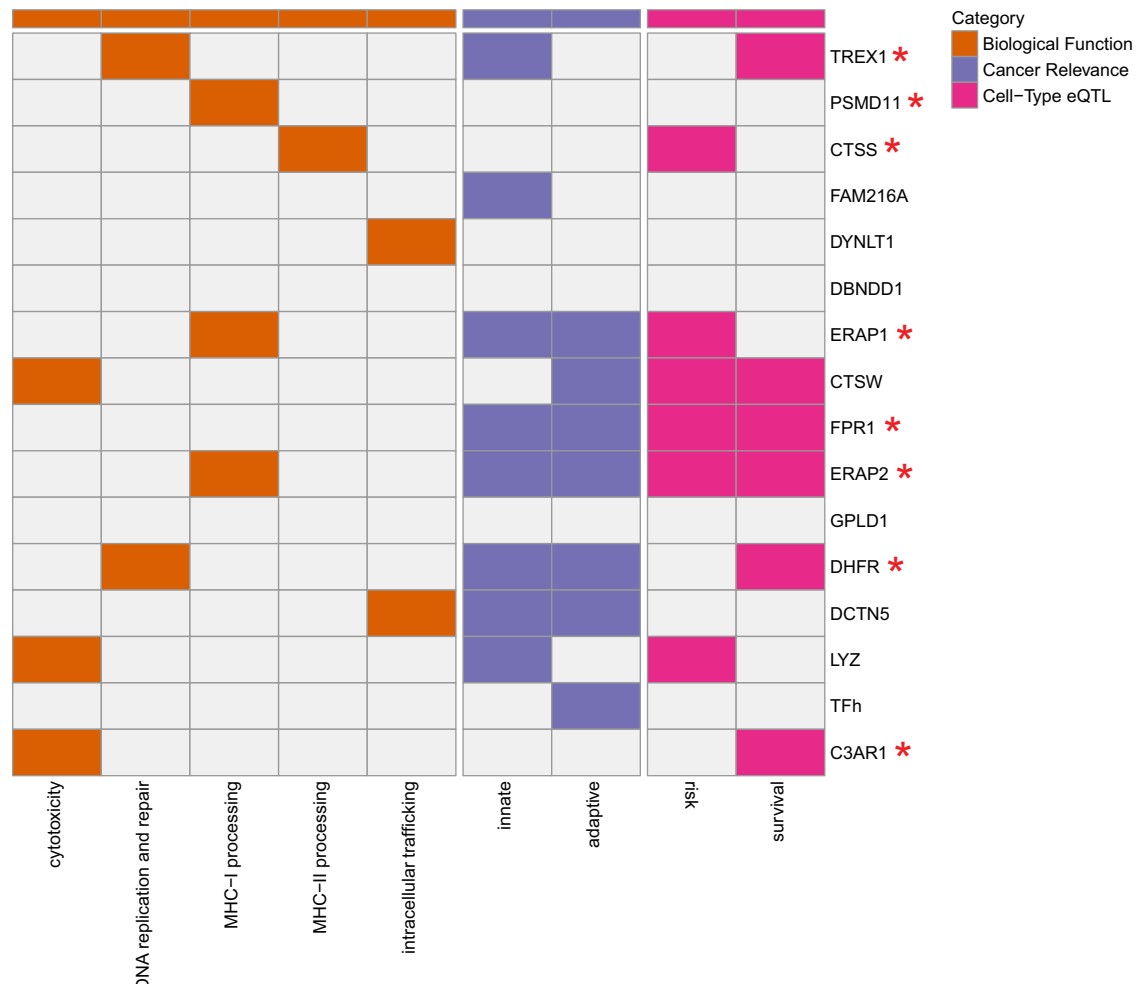

**Fig. 7 | Characterization of genes implicated by PICS model TIME eQTLs.** A map of TIME eQTL biological functions, immune functions and cancer associations for 15 genes implicated as modifiers of immune checkpoint blockade response. Innate immune function indicates that TIME eQTLs are also DICE eQTLs for macrophages, monocytes or dendritic cells. Adaptive immune function indicates that TIME eQTLs are also DICE eQTLs for CD8+ T cells, CD4+ T cells, or B cells. Risk indicates whether a gene was also implicated in PRS models. Survival indicates whether a gene was also implicated in PSS models. Asterisks indicate that a small molecule inhibitor has been reported for a gene. Source data are provided in the Source Data file.

activation and leads to more complete tumor clearance. *CTSS* was both detected and validated as a determinant of risk and response, suggesting MHC Class II could underlie both risk and response to immunotherapy. Together with reports from multiple immune vaccine studies that responses were primarily driven by CD4+ T cells[105–108], these findings place further emphasis on the central importance of MHC II for effective anti-tumor immune responses.

In contrast, a subset of innate immune genes were implicated only in the context of cancer progression and immunotherapy response. Innate immunity acts as the body's first line of defense against microbial pathogens and cancer cells, and involves cells originating in the bone marrow that carry non-polymorphic receptors. Cells of the innate branch such as macrophages and dendritic cells play a pivotal role in the tumor microenvironment creating a hostile pro-inflammatory environment, suppressing T cells, promoting angiogenesis, and initiating lymphangiogenesis. While innate eQTLs such as *FPR1*, *OAS1*, and *LYZ* were also implicated in risk, aspects of immune stimulation related to pathogen and damage associated molecular patterns such as *TREX1* were depleted in risk association. That these genes do not appear in association with risk could indicate that the tumor microenvironment has to reach a certain stage before they are relevant. Involvement of innate immune genes in later disease stages could also potentially indicate a role for certain microbes and

pathogens in prognosis and immunotherapy response that is modified by inter-individual differences in innate immune function. Indeed, it is increasingly appreciated that microbes occupy the tumor niche[109] and can contribute to differences in outcome[110–112].

Immune checkpoint blockade-specific polygenic scores (PICS) derived from TIME eQTLs implicated putative targets to modify anti-tumor immunity; *TREX1* has previously been highlighted as a promising target[73], and small molecule inhibition of CTSS resulted in slower tumor growth and longer survival of mice, with effects comparable to anti-PD-1. CTSS has been reported to affect macrophage function and gene expression levels in autoimmune disease[113] and has reported immune suppressive roles in follicular lymphoma[76,100,114]. We found that inhibition of this gene relieves immune suppression in solid tumors and synergizes with immune checkpoint inhibitors. In solid tumors, reports have highlighted that CTSS can impact TGFβ-related activities[115], autophagy[116] and BRCA1 stability[117], so it is possible that the effects of inhibiting CTSS are not exclusive to the tumor immune microenvironment. Nonetheless, we observed remodeling of the suppressive and inflammatory-like macrophage populations in the mouse tumors treated with CTSS inhibitor.

Notably PICS reproducibly correlated with ICB response across multiple cohorts with melanoma, non-small cell lung cancer (NSCLC) and kidney cancer (RCC). Furthermore, PICS compared favorably with

other popular measures such as tumor mutation burden (TMB) and checkpoint gene expression for predicting binary response category. In RCC the link between tumor mutation burden (TMB) and ICB response is not clear, in contrast to high TMB diseases like melanoma and NSCLC where higher TMB is associated with better responses[61,118–120]. Possibly, in a setting with low TMB such as in RCC, host genetics have more value as prognostic biomarkers. In the future, germline determinants of the TIME could be integrated into predictors alongside other characteristics of the TIME found to inform immune response such as TMB, PD-L1 positivity, the number and quality of T cells[121], IFN-γ response, cytotoxicity scores, T cell activation and T cell exhaustion signatures[59–61,122–129]. Some of these factors require profiling of tumor RNA which is less commonly performed in clinical settings. If germline variants could serve as a proxy for characteristics of the TIME that otherwise require more complex molecular profiling, they could provide an avenue for more cost effective tools for the clinic.

Our analysis had several limitations. We focused on common germline variation; however, rare germline variants have potential to modify the tumor immune microenvironment. In Sayaman et al., MMR rare variants were associated with higher lymphocyte infiltration and BRCA1 mutations with IFN and MHC response modules[52]. Exploration into rare variants in immune genes could reveal aspects of TIME but might also share mechanisms with increased infection rates or immunodeficiencies[130,131]. These individuals may be affected by rare cancer types as observed in transplant and HIV-infected patients[132]. Common SNP to gene linkages were assumed based on SNP association with gene expression. However, it is possible that some eQTLs may be incorrectly linked to target genes or may affect the expression of multiple genes.

Furthermore, our approach prioritized pan-cancer associations, which has the potential advantage of revealing more generalizable associations at the cost of missing cancer-specific effects. Our approach is dependent on the availability of paired genomic and transcriptomic data from tumors, which is currently available only for a few cohorts. Effect sizes associating genetic variants with cellular phenotypes are likely to be larger than those linking genetic variants to diseases[133–135], however the number of associations detected may still be limited by available sample sizes and the limited population diversity thereof. Phenotypes comprising multiple genes are likely to have higher polymorphicity, which could make detection of associations with composite phenotypes such expression-based estimates of pathway activity or immune cell infiltrates more difficult. We were able to impute a subset of our SNPs into existing immune checkpoint blockade study cohorts that had only exome sequencing, but others falling outside of exonic regions could not be analyzed in this context. Studies focused on tumor exomes and transcriptomes could include genome-wide SNP profiling via arrays or low pass whole genome sequencing to allow more effective integration into future studies of germline genomic variation.

## Methods
### Ethics statement
This research was conducted in accordance with the guidelines of the University of California San Diego (UCSD) Institutional Review Board (IRB). The UCSD IRB has determined this study does not involve human subjects research as defined by federal regulations at 45 CFR 46 as it entails secondary analysis of deidentified human data. We have taken all necessary steps to ensure that the study was conducted ethically and in compliance with all relevant guidelines and regulations. All the animal studies were approved by the Institutional Animal Care and Use Committee (IACUC) of university of California, San Diego, with protocol ASP #S15195. All experiments adhere with all relevant ethical regulations for animal testing and research.

### TCGA subject details
The Cancer Genome Atlas (TCGA) consists of tumor and matched normal samples for over 11,000 patients. The Genomic Data Commons (GDC) legacy archive contains germline data for 11,542 samples from 10,875 unique individuals. Samples with TCGA project IDs: DLBC, LAML, and THYM were excluded as they represent cancers derived from immune cells. Pairs of individuals with estimated KING kinship coefficient > 0.177, which represents first-degree relatedness were excluded. TCGA individuals were consented for general research use and no attempts were made to reidentify or contact subjects. Both females and males were included, and sex and individual age were included as covariates. Experiments were not blinded and randomization of subjects was not relevant to the study.

### TCGA genotype processing
Normal (non-tumor) level 2 genotype calls generated from Affymetrix SNP6.0 array intensities using BIRDSUITE (RRID: SCR_001794) software[136] were retrieved from TCGA GDC Legacy Portal (accession date: 04/26/2019) using gdc-client v1.6.0. In these files, each of 906600 SNPs was annotated with an allele count (0 = AA, 1 = AB, 2 = BB, and −1 = missing) and confidence score between 0 and 1. Genotypes with a score larger than 0.1 (error rate >10%) were set to missing and data were reformatted for PLINK (RRID:SCR_001757)[35]. We discarded 322 SNPs with probe names that did not match the hg19 UCSC Genome Browser (RRID:SCR_005780) Affymetrix track (track: SNP/CNV Arrays, table:snpArrayAffy6). Allele counts were converted to alleles using the definitions in metadata distributed with Affymetrix SNP 6.0 Array Documentation and negative strand genotypes were flipped to the positive strand using PLINK.

Pre-imputation processing of autosomal and X chromosome genotypes consisted of the following steps:
1. SNPs with call rate <90% were removed.
2. SNPs with minor allele frequency (MAF) < 1% were removed.
3. Individuals with genotype coverage <90% were removed.
4. Individuals with conflicting gender assignments were flagged.
5. Heterozygous haploid SNPs were set to missing.

After applying these filters, the remaining 800,644 autosomal and 32,809 X chromosome SNPs were input to the secure Michigan Imputation Server[137]. SNPs were imputed with Minimac3/Minimac4 and European HRC Version r1.1 2016 reference with Eaglev2.3 phasing.

Post-imputation processing of genotypes included:
1. SNPs with MAF < 1% were removed.
2. Autosomal SNPs with Hardy–Weinberg Equilibrium <1e-9 were removed.
3. Individuals with high heterozygosity rates (>3 SDs of mean) were removed.
4. Pairs of individuals with kinship coefficient >0.177 (first-degree relatedness) were removed.

Rsq values from INFO files were extracted to annotate genotyping quality. The final genotyping data included 8217 individuals and 7,884,718 variants. Only single nucleotide polymorphisms (SNPs) were analyzed.

### TCGA population stratification
Ancestry filtering was applied using two techniques: (1) k-means clustering and (2) outlier identification. HapMap Phase III genotypes were obtained from the NCBI HapMap ftp site and lifted to hg19 using lift-Over (downloaded 07-09-2019)[138]. Hapmap and TCGA were merged and reduced to a set of 33,675 independent SNPs determined previously through linkage-based filtering using PLINK[135,138]. Pairwise identity-by-state (IBS) between all individuals was calculated and the resulting IBS matrix was used for PCA analysis. Ancestral clusters

were determined by first training k-means clustering using sklearn v0.20.3 on HAPMAP Phase III individuals and then predicted groups in TCGA. TCGA Individuals were grouped into the following HAPMAP groups: (1) TSI, CEU, (2) JPT, CHD, CHB, (3) MEX, (4) GIH, (5) MKK, and (6) YRI, ASW, LWK. Cluster (1) was identified as European individuals.

We ran the aberrant R package v1.0 with lambda 20 for outlier identification[139]. Intersection of k-means clustered individuals and non-outlier individuals from outlier identification analysis was used for the European ancestry discovery cohort in TCGA.

## TCGA phenotype data

PanCanAtlas RNA data from GDC PanCanAtlas Publications Supplemental Data (https://gdc.cancer.gov/about-data/publications/pancanatlas) was downloaded (access date: 10/14/19). Only primary tumors (barcode: 01A/01B/01C) were considered in our analysis. Corresponding clinical metadata were obtained from the GDC Portal (https://tcga-data.nci.nih.gov/docs/publications/tcga/).

The following phenotypes were extracted or generated from RNA-seq data:

1. Immunomodulators: 436 genes used to define immune states from Thorsson et al.[44].
2. Immune checkpoint molecules: 78 immune checkpoint stimulatory and inhibitory molecules from Thorsson et al.[44].
3. Antigen presentation: 231 antigen presentation genes from Gene Ontology [GO_REF:0000022].
4. Immune cell markers: 60 immune cell type markers from Danaher et al.[140].
5. *IFN-γ*: *IFN-γ* genes retrieved from Biocarta [Systematic Name: M18933].
6. *TGF-β*: *TGF-β* genes retrieved from Biocarta [Systematic Name: M22085].
7. Immune states: Individual level scores for 6 immune states [wound healing, *IFN-γ* dominant, inflammatory, lymphocyte depleted, immunologically quiet, and *TGF-β* dominant] from Thorsson et al.[44].
8. Immune infiltration levels: 22 relative immune infiltration estimates from CIBERSORTx[141] using the LM22 signature matrix.

Phenotypes with greater than 10% zero values were excluded and rank-based inverse normal transformation (Supplementary Fig. 1) was applied to each tumor type using Eq. 1[142]. This transformation causes each phenotype to have an identical distribution in each tumor type, which removes tumor-type specific information.

$$qnorm((rank(x, na.last = ''keep'') - 0.5)/sum(!is.na(x))) \quad (1)$$

A total of 733 phenotypes remained for preliminary analyses.

For HLA allele-specific expression, TCGA tumor-specific RNA BAM files were downloaded from the GDC on 07/16/2019. The HLApers[143] kallisto-based pipeline was used with gencode v30 annotations[144]. Default parameters were used and the two alleles with the highest calculated expression were retained for each HLA gene if there were more than 2 alleles reported. The top 2 highest expressed HLA alleles for each gene were averaged for input into SNP analyses. If expression for at least two alleles was not calculated, expression was set as missing for the sample. Only primary samples (01A/01B/01B) were considered for analysis. Summed HLA allele-specific expression was inverse-rank normalized by cancer type and used for downstream analyses.

## TCGA GCTA analysis

SNP heritability estimates were calculated with the genomic-relatedness-based restricted maximum-likelihood (GREML) approach implemented in GCTA (Genome-wide Complex Trait Analysis) v1.93.2beta[145,146]. Genetic relationship matrices (GRMs) which measure genetic similarity of unrelated individuals (GRM < 0.05) were constructed for the autosomal and X chromosomes for the European cohort. Benjamini-Hochberg false discovery rates (FDR) were calculated using statsmodels[147]. Immune traits were considered sufficiently heritable if the V(g)/V(p) value was >0.05 using the full GRM.

As highly polymorphic regions such as HLA and KIR gene regions can inflate heritability estimates, we conducted a 2-state GCTA analysis with separate GRMs for HLA/KIR regions (HLA chr6:28,477,797-33,448,354, KIR chr19:55,228,188-55,383,188) and with the rest of the genome excluding HLA/KIR regions. Age and sex were included as covariates. An FDR < 0.05 was used to identify SNP-heritable IP components from 2-state analysis. If an IP component had high SNP heritability using the HLA/KIR GRM, a conditional GWAS analysis was conducted; otherwise, a standard GWAS analysis with Bonferroni-corrected suggestive p-value threshold was conducted. Ultimately, 140 IP components outside of the HLA/KIR regions and 17 IP components within the HLA/KIR regions were identified. We repeated the 2-state analysis for breast cancer only samples using age, ER, PR, and HER2 status as a covariate. Hormone receptor status was categorical and retrieved from clinical files describing IHC results.

## TCGA phenotype principal component analysis

In all, 157 SNP-heritable components were analyzed using sklearn. IP component values were scaled by Sklearn Standard Scaler and used for principal component analysis (PCA). Ordinary least squares (OLS) regression was performed with 157 IP components and principal components, wherein the beta coefficient represents the degree of change in principal component for every unit change in IP component. P-values indicate whether a coefficient was significantly different from 0.

## TCGA GWAS analysis

The GLM method in PLINK was used to conduct association analyses with IP components. All associations were adjusted for covariates of age, sex, and the first ten principal components. Gene expression values, CIBERSORTx relative infiltration estimates, and immune state scores were inverse-rank normalized by tissue type to control for tissue-type expression effects. Significant associations were identified with the PLINK clumping method using the primary suggestive threshold corrected for the number of phenotypes tested[148] ($1 \times 10^{-5}/140$) using a kb threshold of 500, and an $R^2$ threshold of 0.5.

To determine if variants had been implicated in previous cancer GWAS studies, variants were input into the LDlink server (https://ldlink.nci.nih.gov/?tab=ldtrait) using parameters "EUR" population, an $R^2$ threshold of 0.5 and base pair window of 500kb[54,55]. We also retrieved the Vanderbilt PheWAS catalog[56] and any TIME eQTLs in high linkage disequilibrium ($R^2 > 0.5$) with Vanderbilt PheWAS catalog cancer risk TIME eQTLs were included as cancer risk variants. Lastly, we assessed TIME eQTLs by PheWAS analysis in the UK Biobank (detailed below).

## TCGA conditional HLA analysis

The PLINK GLM method was used to run stepwise conditional analysis for identification of independent HLA associations[39]. The most significant initial associations detected with HLA region phenotypes by standard GWAS analysis were incorporated as covariates in the subsequent round. Specifically, we re-ran the analysis with chromosome 6 variants including the most significant SNP (lowest p-value in the previous round) as a covariate. Analysis was conducted until no SNPs with Bonferroni-corrected $p$-value < ($1 \times 10^{-5}/17$) remained. Analysis for HLA-DRB5 was revisited using only individuals with HLA-DRB1*15 and HLA-DRB1*16 allele calls indicating haplotypes where the HLA-DRB5 gene is present. We re-ran conditional GWAS analysis only within individuals with these alleles ($n = 1564$). SNPs with Bonferroni-corrected $p$-value ($p < 1 \times 10^{-5}/17$) were kept for further analysis.

## Literature TIME Associations

We compiled existing germline variants associated with the tumor immune microenvironment (TIME) or ICB response from the literature. We collected 14 studies with their descriptions below:

1. Kogan et al. (2018): Discovery of *FGFR4* germline variant which enhances *STAT3* activity impeding CD8 T cell infiltration.
2. Queirolo et al. (2017): Investigation of 6 *CTLA-4* SNVs in 173 metastatic melanoma patients with overall response and survival information.
3. Uccellini et al. (2012): *IRF5* polymorphism was associated with non-response to adoptive therapy with TILs.
4. Bedognetti et al. (2013): *CXCR3* and *CCR5* genetic polymorphisms were evaluated for expression of respective ligands and TIL migration.
5. Lim et al. (2018). Systematic identification of germline genetic polymorphisms associated xCell cell type gene signatures (gsQTLs) in TCGA.
6. Shahamatdar et al. (2020). Systematic identification of germline genetic polymorphisms associated with immune infiltration in TCGA.
7. Ostendorf et al. (2020). Identification of *APOE2* and *APOE4* germline variants associated with melanoma progression and ICB response in mice.
8. Zhang et al. (2019). Identification of breast-cancer-associated variant modulating *CTSW* expression.
9. Sayaman et al. (2020). Systematic identification of germline variants associated with 33 immune traits including leukocyte subsets, adaptive receptor, immune expression signatures.
10. Yoshida et al. (2021). Identification of 2 *PD-L1* variants associated with survival outcomes in advanced non-small-cell lung cancer patients.
11. Kula et al. (2020). Review of 10 *PD-L1* genetics variants.
12. Salmaninejad et al. (2018). Review of 5 frequently studied *PD-1* genetic variants.
13. Sasaki et al. (2014). Characterization of *PD-1* promoter variant and association with survival in non-small cell lung cancer.
14. Tang et al. (2015). Characterization of 3 *PD-1* variants and association with cancer risk.

For Sayaman et al., 598 significant associations were identified, 520 of which were within the MHC II region. To identify independent Sayaman et al SNPs, we performed linkage disequilibrium based clumping with the same parameters used for our analysis. After clumping, 55 independent Sayaman et al SNPs remained.

## TIME eQTL Annotation

Variants were annotated with VEP (Variant Effect Predictor)[149] with default parameters and the GRCh37 reference genome. Coding variants were mapped to protein sequences using the Uniprot GFF file.

GREGOR (RRID: SCR_009165) was used to analyze SNP enrichment at epigenetic features. We obtained 479 bed files for 11 histone experiments and 52 cell types from ENCODE (RRID:SCR_015482) (downloaded on 3 May 2020). Only "stable peaks" and "replicated peaks" files were kept for analysis. If more than 1 bed file for a cell type and transcription factor were available, the files were combined, resulting in 259 files.

GREGOR was run with EUR Reference files made from the 1000 Genomes Project data with an LD window size of 1MB and LD $R^2 > 0.7$. Enrichment ratios were calculated by taking the difference between observed and expected number of SNPs and dividing by the expected number of SNPs. Any files with Audit errors were excluded.

## Cell-type eQTL analysis

We followed the GTEx approach for cell type interaction eQTL discovery[36]. We ran a linear regression model with an interaction term accounting for interactions between genotype and cell type enrichment from xCell[150] Eq. 2:

$$p \sim g + i + g^o i + C \qquad (2)$$

where p is the IP component vector, g is the genotype vector, i is the inverse normal transformed by tissue type xCell enrichment score[150], and the interaction term g ∘ i corresponds to pointwise multiplication of genotypes and cell type enrichment scores. The same covariates, denoted by C, were used as in the regular immune microenvironment GWAS analysis. Benjamini-Hochberg FDR was calculated for the beta coefficient of the interaction term and variants with FDR < 0.05 were identified as significant.

DICE expression quantitative trait loci (eQTLs) were obtained at https://dice-database.org/. Methods associated with DICE eQTL discovery are published in Schmiedel et al.[27].

## Non-linear polygenic score construction

Using the approach outlined in Elgart et al.[65], we generated three distinct polygenic scores to characterize TIME eQTLs as predictive of risk, survival, or ICB response. For each predictive task, we built models using a training cohort and evaluated them on a held-out validation cohort that was independent of the training cohort when available. First, we conducted three separate association analyses to determine the effect of each TIME eQTL on each outcome, including only individuals in the respective training cohorts to calculate beta and significance values and controlling for covariates relevant to each outcome. Next, nominally significant eQTLs from these associations were subjected to shrinkage-based selection using LASSO[151]. We tuned the parameter controlling the strength of shrinkage (α) in the LASSO by testing a range of α's for each model, from those that removed all eQTLs under consideration to those that kept all of them, and chose the one that maximized AUC ROC on the training cohort. The eQTLs that passed this selection process were used as features to construct an XGBoost[152] model predictive of the outcome of interest. We only fit XGBoost models on the respective training cohorts and then applied the models to calculate scores on the validation cohorts. We also performed a feature importance analysis for each model by using the model.feature_importances_ function from the python xgboost package (version 1.6.2). XGBoost model parameters were set to default and a random seed was fixed across all analyses to ensure reproducibility.

Polygenic Risk Scores (PRS) for melanoma and prostate cancer were constructed from TIME eQTLs with nominal cancer risk associations based on our UK Biobank PheWAS. Beta values were extracted from the UK Biobank PheWAS with cancer ICD10 codes. The melanoma risk model (number of SNPs=43) was validated using the Geneva melanoma cohort (excluding individuals with no FH of melanoma), while the prostate cancer risk model (number of SNPs=26) was validated on all individuals in the ELLIPSE prostate cancer cohort. PRS quantiles and corresponding odds ratios were presented.

Polygenic survival scores (PSS) were constructed for cancer types with available stage information and at least 100 samples. This resulted in 15 cancer types for analysis. We constructed PSS based on TIME eQTLs nominally associated with OS and PFS in cancer-specific Kaplan-Meier analyses (*P* < 0.05). Cancer-type specific beta values for each SNP were obtained from a Cox Proportional Hazards model measuring contribution to survival outcomes while adjusting for relevant tumor type-specific covariates (Supplementary Data 13). TCGA cohorts were split 70:30 into train and validation partitions. The PSS model for TCGA LUAD (number of SNPs=28) was validated in the Sherlock cohort. Kaplan-Meier curves were generated for the 30% of held-out TCGA-LUAD samples not used for model training, and all individuals in the independent Sherlock validation sets based on quartile stratification (low, middle, middle, high) and significance was assessed through

logrank tests between low and middle, low and high and middle and high.

Polygenic ICB scores (PICS) were constructed from nominally significant TIME eQTLs identified in the METAL analysis of response (iRecist: CR, PR, SD) across four ICB-treated melanoma cohorts (Van Allen, Hugo, Riaz, and Snyder). The PICS model (number of SNPs = 31) was validated on two independent ICB-treated cohorts (Rizvi and Miao). ROC-AUC and Mann-Whitney U tests[153] were the primary evaluation metrics used to assess PICS performance for predicting ICB response. We further conducted ROC-AUC analysis with clinical variables (age, sex) alone, PICS alone, and PICS with clinical variables. Logistic regression was used to estimate the variance in response status explained by PICS, TMB and checkpoint gene expression. McFadden pseudo-R2 was reported and models were compared by anova.

### Risk analysis - UK Biobank

To assess cancer risk, we conducted PheWAS with cancer ICD10 codes in the UK Biobank. UK Biobank subjects were subsetted into separate ethnic-racial groups following continental ancestry prior to analysis. To identify the European-ancestry samples, we started with directly called genotype data and identified a set of overlapping SNPs with 1000 Genomes Project and AWS (RRID:SCR_008801) (1KG) population and then merged them together. Next, we pruned the SNP set so remaining SNPs were in linkage equilibrium using PLINK[35]. flashpca was used to calculate principal components for 1KG SNPs[154]. The UK Biobank samples were projected onto 1KG space using flashpca. To identify subjects of European ancestry, we utilized Aberrant to generate clusters with a broad set of lambda values (clustering thresholds) and checked that the cluster included all 1KG subjects of European ancestry and maximized the total number of UK Biobank subjects (lambda = 8.2)[139]. Finally, we compared the self-reported race/ethnicity of subjects within this cluster and removed samples that were discordant. We identified 454,487 subjects of European ancestry. To identify the unrelated samples from the finalized European list, we used the relatedness file provided by UK Biobank and a custom script was used to select unrelated samples while maximizing sample counts. The final European unrelated set included 382,841 subjects. Variant dosages extracted from imputed UK Biobank BGEN files were used for PheWAS analysis with PLATO v2.0.0[155].

ICD10 diagnosis codes associated with neoplasms and immune disorders were collapsed according to level-1 groupings used by UK Biobank resulting in a total of 24 groups. For example, C00-C14 is one of the groups containing ICD10 codes associated with malignant neoplasm of lip, oral cavity, and pharynx. Individuals with diagnosis code in a group were coded as 1, with the remaining individuals coded as 0. Logistic regression was conducted with UK Biobank binary files containing HLA-immune variants, logistic phenotype file, and age, sex, and principal components 1-10 as covariates. *P* values were Benjamini–Hochberg FDR adjusted.

### TCGA survival analysis

Kaplan-Meier analysis of immune microenvironment associations were conducted with overall and progression-free survival retrieved from Liu et al.[156] by cancer type using the lifelines package v0.25.11. As recommended by Liu et al., TCGA cancer types, TGCT and PCPG, were excluded as survival data did not meet quality standards. TCGA individuals were divided into three groups based on genotype calls: minor allele homozygotes, heterozygotes and major allele homozygotes. Significance was determined using the logrank test between minor allele and major allele homozygotes. Only SNPs with at least 1% minor allele frequency in each cancer type and more than 1 minor allele homozygous individual were considered for analysis. Only variants with a nominal $p < 0.05$ were considered as candidate features for PSS model construction.

### High Density Melanoma Cohort and ELLIPSE Consortium genotypes

Raw genotypes for the High Density Melanoma Cohort and the ELLIPSE Consortium were downloaded from dbgap under accession phs000187.v1.p1[69,157]. Duplicate genotypes were removed and lifted over to the hg19 reference genome. SNPs with call rate <90% and minor allele frequency (MAF) < 1% were removed. Individuals with genotype coverage <90% were removed. Using snpflip, variants were flipped such that they were oriented to the "+" strand. 822,808 variants and 3033 individuals remained for genotype imputation by Michigan Imputation Server[137] (Minimac3/Minimac4, European HRC Version r1.1 2016 reference, Eaglev2.3 phasing).

Raw genotypes for ELLIPSE Consortium were downloaded from dbgap under accession phs001120.v2.p2[158]. PLINK genotype files consisting of 505,219 calls from the following consent groups were compiled: c1-c3,c6,c8,c10-18,c20,c23,c25,c27-28. Pre-imputation processing of autosomal and X chromosome genotypes followed below steps:

1. Duplicated variants were removed.
2. Heterozygous haploid SNPs were set to missing.
3. SNPs with call rate <90% were removed.
4. SNPs with minor allele frequency (MAF) < 1% were removed.
5. Individuals with genotype coverage <90% were removed.
6. Non-ACGT variants were removed.

Strand flips were reversed using snpflip. After preprocessing genotypes, the remaining 410,116 SNPs and 91,644 individuals were input to the secure Michigan Imputation Server (RRID:SCR_017579)[137]. Whole-genome SNPs were imputed with Minimac4 (RRID:SCR_009292) and ancestry-matched reference panel 1000 Genomes Project Phase 3 version 5 (RRID:SCR_008801). Finally, post-imputation duplicated SNPs and SNPs with MAF < 1% were removed.

Necessary PRS TIME eQTLs were extracted from imputed genotypes.

### SHERLOCK genotypes

Sherlock genotype processing are detailed in original publication[71]. Briefly, germline DNA from 256 individuals were obtained. 24 were excluded due to either quality control issues or computational artifacts, resulting in 232 samples. Variants were called with GATK Haplotyper algorithm[159]. Final calls were annotated with ANNOVAR[160].

### Immunotherapy response analysis

Raw fastq files were obtained using SRA toolkit v2.9.6-1-ubuntu64 for the following immune checkpoint trials: Hugo et al. 2016 (SRA accession: SRP090294, SRP067938; Cancer: melanoma)[57], Van Allen et al. (SRA accession: SRP011540, Cancer: melanoma)[58], Miao et al. (SRA accession: SRP128156, Cancer: clear cell renal carcinoma)[161], Riaz et al. (SRA accession: SRP095809, SRP094781; Cancer: melanoma)[60], Rizvi et al. (SRA accession: SRP064805, Cancer: non-small cell lung cancer)[59], Snyder et al. (SRA accession: SRP072934, Cancer: melanoma)[61]. Reads were aligned to UCSC hg19 coordinates using BWA (RRID:SCR_010910) v0.7.17-r1188[162]. Reads were sorted by SAMTOOLS (RRID:SCR_002105) v0.1.19[163,164], marked for duplicates with Picard Tools (RRID:SCR_006525) v2.12.3 and recalibrated with GATK (RRID:SCR_001876) v3.8-1-0[165–167]. Germline variants were called from sorted BAM files using DeepVariant v0.10.0-gpu[168,169]. The final immunotherapy cohort consisted of 68 clear cell renal carcinoma, 279 melanoma and 34 non-small-cell lung cancer patients.

To evaluate the quality of SNP imputation from whole exome data, we took advantage of the TCGA having both. Of the 1,322,586 variants available from DeepVariant analysis of immunotherapy cohort, 225,000 were available in TCGA imputed data. We extracted these 225,000 variants from TCGA and input into the Michigan Imputation Server (reference panel: HRC, phasing: Eagle). We

compared genotypes from whole-exome calls vs. original Affymetrix-based TIME-SNP calls. Variants with >5% mismatches in genotype calls, minor allele frequency <5% in any cohort or imputation accuracy ($R^2 < 0.3$) were excluded. Only variants with at least 5% frequency in all 4 melanoma cohorts used for discovery analysis were considered for ICB analysis, leaving 525 SNPs.

Population stratification analysis was conducted by taking overlapping variants between TCGA and ICB cohorts. Variants with MAF differences >0.1% were excluded resulting in 3612 frequency-concordant variants. PLINK IBD analysis was conducted and top 10 principal components were included in association analysis.

Subject phenotypes were downloaded from supplementary information of ICB trial publications. Four melanoma cohorts were used as the discovery cohort for ICB-associated variants, while Miao et al. renal cell carcinoma and Rizvi et al. non-small cell lung cancer cohorts were used for validation. Response phenotypes were determined from iRECIST criteria[170]. Patients were categorized as responders if they had iRECIST criteria: CR (complete response), PR (partial response), and SD (stable disease). Non-responders had iRECIST criteria: PD (progressive disease). This resulted in 114 responders and 165 non-responders. Genome-wide association studies (GWASs) were conducted for ICB responders within each ICB-cohort using PLINK. Age, sex, and the top 10 principal components were included in the logistic analysis as covariates. We then used METAL (version release 2011-03-25)[64] with a sample size weighting scheme to perform a pan-study melanoma meta-analysis for ICB response. Only variants with a nominal METAL analysis p < 0.05 were considered as candidate features for PICS model construction.

### Immune checkpoint blockade response RNA-seq
FASTQ/BAM files were downloaded for 33 RCC and 120 melanoma patients. BAM files were converted to FASTQ using bam2fq[164]. Unpaired reads were removed using fastq pair[171]. Paired reads were aligned with STAR (RRID:SCR_004463) v2.4.1d[172] to GRCh37 reference alignment. RSEM v1.2.21[173] was used for transcript quantification. TPM values were log2 transformed for analyses. Differential gene expression analysis between responders and non-responders from cohorts Riaz et al.[60], Hugo et al. 2016, Miao et al.[161], and Van Allen et al.[58] was performed using the DESeq2[143] package in R. Cohort was included as a covariate when calculating top differentially expressed genes.

### Mouse experiments
Wild-type C57BL/6 (RRID:IMSR_JAX:000664) were purchased from The Jackson Laboratory. Mice at Moores Cancer Center, UCSD are housed in micro-isolator and individually ventilated cages supplied with acidified water and fed 5053 Irradiated Picolab Rodent Diet 20 lab diet. Temperature for laboratory mice in our facility is mandated to be between 65 and 75 °F (~18–23 °C) with 40–60% humidity. All animal manipulation activities are conducted in laminar flow hoods. All personnel are required to wear scrubs and/or lab coat, mask, hair net, dedicated shoes, and disposable gloves upon entering the animal rooms. A 12 light/12 dark cycle was used for the mice. In all, $2 \times 10^5$ MC38 (RRID:CVCL_B288) cells were transplanted into the flank of 8–10 female C57Bl/6 (RRID:IMSR_JAX:000664) mice, aged 7-8 weeks. Where indicated, when tumors reached 100 mm³, mice were randomized and treated with anti-PD-1 (10 mg/kg i.p., Bio X Cell Cat# BE0146, RRID: AB10949053, clone RMP1-14), CTSS inhibitor (5 mg/kg, i.p., APExBio) or isotype control antibody (Bio X Cell, Cat #BE0091). Treatments were given 3 times a week. Mice were euthanized per ASP guidelines when tumors reached 1500 mm³ or when control mice succumbed to tumor burdens, and tumors were taken for flow cytometric analysis. All mice were euthanized by trained personnel with carbon dioxide inhalation in a euthanasia chamber. Cervical dislocations were used as a secondary means to assure death after euthanasia with CO2. MC38 cells were not screened using STR profiled on site.

### Flow cytometry
For in vivo studies, tumors were dissected, minced, and re-suspended in complete media (DMEM with 10% FBS and 1% antibiotics) supplemented with Collagenase-D (1 mg/mL; Roche) and incubated at 37 °C for 30 min with shaking to form a single-cell suspension. Tissue suspensions were washed with fresh media and passed through a 70-μm strainer. Cells were stained for viability with Zombie Aqua Viability Dye (BioLegend) according to manufacturer's instructions. Cell surface staining was done for 30 min at 4 °C with the following antibodies: Live/Dead Fixable Aqua stain (1:1000), CD11b-BV711 (M1/70) (1:200), CD68-APC/Cy7 (FA-11) (1:100), F4/80-PE/Dazzle (BM8) (1:200), I-A/I-E (M5/114.15.2) (1:200), and Arginase 1 (A1exF5) (1:100). All antibodies were purchased from BioLegend, and the viability stain and Arginase 1 was purchased from ThermoFisher Scientific. The gating strategy for M1 and M2 macrophages are shown in Supplementary Fig. 8.

### RT-PCR
RNA from MC38 (RRID:CVCL_B288) tumors was extracted using the RNeasy Mini Kit (Qiagen catalog #74104). 500 ng of RNA per reaction was used to prepare cDNA with the SuperScript™ VILO™ cDNA Synthesis Kit (ThermoFisher Scientific) following manufacturer's instructions. The cDNA was used to set up the RT-PCR reaction with 4 technical replicates per tumor with the Fast SYBR™ Green Master Mix (ThermoFisher Scientific) according to manufacturer's instructions. PCR quantification was conducted using the $2^{-\Delta\Delta CT}$ method and normalized to the housekeeping gene β-actin. Primers used for *CTSS* expression quantification are detailed in Supplementary Data 19.

RNA-seq and CIBERSORTx infiltration estimates for M4 melanoma mouse model were obtained from GEO accession (GSE144946). Responders were mice whose size at harvest was smaller than the last dose of anti-CTLA-4. RNA-seq counts were converted to TPM and log2 normalized.

### Reporting summary
Further information on research design is available in the Nature Portfolio Reporting Summary linked to this article.

## Data availability
All human data used in this study come from publicly available sources, however some of these sources require controlled access. The raw data can be obtained directly from the source studies. The processed form of the data used to support the findings of this study are available on request from the corresponding authors HC and MP. Because many of the sources are controlled access, the requestor must have approved access for the data to be shared.

For Data Access to processed genotyping and transcriptomic data, contact corresponding authors with proof of access to dbGaP studies: TCGA[174] (dbgap accession: phs000178.v11.p8); UK Biobank[175] [https://www.ukbiobank.ac.uk/enable-your-research/apply-for-access]; Hugo et al. 2016[57] (SRA accession: SRP090294, SRP067938); Van Allen et al.[58] (dbgap accession: phs000452.v3.p1, SRA accession: SRP011540); Miao et al.[161] (dbgap accession: phs001493.v2.p1, SRA accession: SRP128156); Riaz et al.[60] (SRA accession: SRP095809, SRP094781); Rizvi et al.[59] (dbgap accession: phs000980.v1.p1, SRA accession: SRP064805); Snyder et al.[59,61] (dbgap accession: phs001041.v1.p1, SRA accession: SRP072934); Oncoarray Prostate Cancer[176] (dbgap accession: phs001120.v2.p2); High Density Analysis of Melanoma[69,157] (dbgap accession: phs000933.v3.p1).

The remaining data are available within the Source Data file.

## Code availability
All code used for analysis and figure generation are available at https://github.com/cartercompbio/TIMEgermline [177].

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

## Acknowledgements

This work was supported by Emerging Leader Award from The Mark Foundation for Cancer Research, grant #18-022-ELA to H.C., NIH grant R01CA269919 to H.C. and L.B.A., NIH grant R01CA220009 to H.C. and M.Z., U01CA196406 to O.H., NIH grant R24 AI108564 to P.V., NIH grants R01 CA247551 and U01 DE028227 to J.S.G., NIH grants U01CA253547 and U24CA258406 to J.P.M. and 1F30CA247168-01 and T32CA067754 to M.P. This work was supported, in part, by funding from the NIH intramural research program and NCI 2019 FLEX Award.

The results shown here are in large part based upon data generated by the TCGA Research Network: https://www.cancer.gov/tcga and Genotype-Tissue Expression (GTEx) Project: https://gtexportal.org/home/. GTEx was supported by the Common Fund of the Office of the Director of the National Institutes of Health, and by NCI, NHGRI, NHLBI, NIDA, NIMH, and NINDS. The data used for the analyses described in this

manuscript were obtained from the GTEx Portal on 10/10/20. This research has been conducted using the UK Biobank Resource under project ID 37671, supported by NIH grant R00HL122515 to RMS.

For Rizvi et al. non-small-cell lung cancer immunotherapy analysis, we used dbGaP data from accession phs000980.v1.p1. We thank the members of the Thoracic Oncology Service and the Chan and Wolchok labs at MSKCC for helpful discussions. We thank the Immune Monitoring Core at MSKCC, including L. Caro, R. Ramsawak, and Z. Mu, for exceptional support with processing and banking peripheral blood lymphocytes. We thank P. Worrell and E. Brzostowski for help in identifying tumor specimens for analysis. We thank A. Viale for superb technical assistance. We thank D. Philips, M. van Buuren, and M. Toebes for help performing the combinatorial coding screens. This work was supported by the Geoffrey Beene Cancer Research Center (MDH, NAR, TAC, JDW, AS), the Society for Memorial Sloan Kettering Cancer Center (MDH), Lung Cancer Research Foundation (WL), Frederick Adler Chair Fund (TAC), The One Ball Matt Memorial Golf Tournament (EBG), Queen Wilhelmina Cancer Research Award (TNS), The STARR Foundation (TAC, JDW), the Ludwig Trust (JDW), and a Stand Up To Cancer-Cancer Research Institute Cancer Immunology Translational Cancer Research Grant (JDW, TNS, TAC). Stand Up To Cancer is a program of the Entertainment Industry Foundation administered by the American Association for Cancer Research. For Snyder et al. melanoma immunotherapy analysis, we used dbGaP data from accession phs001041.v1.p1. We thank Martin Miller at Memorial Sloan Kettering Cancer Center (MSKCC) for his assistance with the NetMHC server, Agnes Viale and Kety Huberman at the MSKCC Genomics Core, Annamalai Selvakumar and Alice Yeh at the MSKCC HLA typing laboratory for their technical assistance, and John Khoury for assistance in chart review. For Miao et al. renal cell carcinoma immunotherapy analysis, we used dbGap data from accession phs001493.v2.p1. This study was supported by an AACR KureIt grant. Hugo et al. melanoma samples were acquired from SRA using accession numbers SRP067938 and SRP090294. Riaz et al. melanoma samples were acquired from SRA using accession number SRP095809. For Van Allen et al. melanoma sample, data was acquired from dbgap accession phs000452.v2.p1.

ELLIPSE Genotypes were accessed under dbgap accession phs001120.v1.p1. This work was supported by the GAME-ON U19 initiative for prostate cancer (ELLIPSE): U19 CA148537. We would like to acknowledge the NCRN nurses and Consultants for their work in the UKGPCS study. We thank all the patients who took part in this study. This work was supported by Cancer Research UK (grant numbers C5047/A7357, C1287/A10118, C1287/A5260, C5047/A3354, C5047/A10692, C16913/A6135, and C16913/A6835). We would also like to thank the following for funding support: Prostate Research Campaign UK (now Prostate Cancer UK), The Institute of Cancer Research and The Everyman Campaign, The National Cancer Research Network UK, The National Cancer Research Institute (NCRI) UK. We are grateful for support of NIHR funding to the NIHR Biomedical Research Centre at The Institute of Cancer Research and The Royal Marsden NHS Foundation Trust. The MEC was supported by NIH grants CA63464, CA54281 and CA098758. High Density Melanoma Genotypes were accessed under dbgap accession phs000187.v1.p1. Research support to collect data and develop an application to support this project was provided by 3P50CA093459, 5P50CA097007, 5R01ES011740, and 5R01CA133996.

## Author contributions

M.P. and H.C. conceived the work, designed and analyzed the experiments and wrote the paper with assistance from P.V., W.T., M.Z., J.M., S.P., O.H., C.F., G.M., C.P.D., L.B.A., J.A., T.L., T.Z., and A.C.; W.T. and C.F. assisted in statistical analyses; A.C. and G.M. assisted in HLA region analysis; J.V.T. and T.S. assisted in genetic model analysis; B.J.S., C.G.C., P.V., and D.K. assisted in cell-type specificity analysis; S.G. assisted in epigenetic analysis; H.K., S.C., and R.M.S. assisted in UK Biobank Phe-WAS analysis; V.W. and J.S.G. performed MC38 mouse experimental validation; C.P.D., E.G., and G.M. provided M4 mouse validation results; L.B.A., J.A., T.L., and T.Z. assisted in Sherlock validation.

## Competing interests

S.P.P. receives scientific advisory income from: Amgen, AstraZeneca, Bristol-Myers Squibb, Certis, Eli Lilly, Jazz, Genentech, Illumina, Merck, Pfizer, Rakuten, and Tempus. S.P.P.'s university receives research funding from: Amgen, AstraZeneca/MedImmune, Bristol-Myers Squibb, Eli Lilly, Fate Therapeutics, Gilead, Iovance, Merck, Pfizer, Roche/Genentech, and SQZ Biotechnologies. R.M.S. has a service contract with Travere Theraputics. L.B.A. is a compensated consultant and has equity interest in io9, LLC. His spouse is an employee of Biotheranostics, Inc. L.B.A. is also an inventor of a US Patent 10,776,718 for source identification by non-negative matrix factorization. L.B.A. declares U.S. provisional applications with serial numbers: 63/289,601; 63/269,033; 63/366,392; 63/367,846; 63/412,835. J.S.G. reports scientific advisory income from Domain Pharmaceuticals, Pangea Therapeutics, and io9, and is founder of Kadima Pharmaceuticals, all unrelated to the current study. O.H. is a current employee and stockholder of Zentalis pharmaceuticals Inc. M.Z. is a board member of Invectys Inc. All other authors declare that they have no competing interests.

## Additional information

[1]Biomedical Sciences Program, University of California San Diego, La Jolla, CA 92093, USA. [2]Bioinformatics and Systems Biology Program, University of California San Diego, La Jolla, CA 92093, USA. [3]Department of Pharmacology, UCSD Moores Cancer Center, La Jolla, CA 92093, USA. [4]Laboratory of Cancer Biology and Genetics, National Cancer Institute, National Institutes of Health (NIH), Bethesda, MD 20892, USA. [5]Undergraduate Bioengineering Program, Jacobs School of Engineering, University of California San Diego, La Jolla, CA 92093, USA. [6]La Jolla Institute for Immunology, La Jolla, CA 92037, USA. [7]Division of Epidemiology, Herbert Wertheim School of Public Health and Human Longevity Science, University of California San Diego, La Jolla, CA 92093, USA. [8]Canyon Crest Academy, San Diego, CA 92130, USA. [9]Undergraduate Biology and Bioinformatics Program, University of California San Diego, La Jolla, CA 92093, USA. [10]Division of Cancer Epidemiology and Genetics, National Cancer Institute, National Institutes of Health (NIH), Bethesda, MD 20892, USA. [11]Department of Pathology, University of California San Diego, La Jolla, CA 92093, USA. [12]Division of Biomedical Informatics, Department of Medicine, University of California San Diego School of Medicine, La Jolla, CA 92093, USA. [13]Center for Personalized Cancer Therapy, Division of Hematology and Oncology, UC San Diego Moores Cancer Center, San Diego, CA 92037, USA. [14]Department of Cellular and Molecular Medicine, University of California San Diego, La Jolla, CA 92093, USA. [15]Department of Bioengineering, University of California San Diego, La Jolla, CA 92093, USA. [16]Moores Cancer Center, University of California San Diego, La Jolla, CA 92093, USA. [17]Department of Medicine, Division of Medical Genetics, University of California San Diego, La Jolla, CA 92093, USA. [18]The Laboratory of Immunology and Department of Medicine, University of California San Diego, La Jolla, CA 92093, USA. [19]Center for Population Neuroscience and Genetics, Laureate Institute for Brain Research, Tulsa, OK 74136, USA. [20]Department of Radiology, University of California San Diego, La Jolla, CA 92093, USA. [21]Division of Biostatistics, Herbert Wertheim School of Public Health and Human Longevity Science, University of California San Diego, La Jolla, CA 92093, USA. ✉e-mail: hkcarter@health.ucsd.edu

