## [Peer Review File · Nature Communications]

Germline modifiers of the tumor immune microenvironment implicate drivers of cancer risk and immunotherapy responseReviewers' Comments:

Reviewer #1:

Remarks to the Author:

The extent and characterization of germline genetic contribution to cancer immune responsiveness remains largely uncharacterized.

So far three large genome-wide analyses have been performed using the TCGA dataset which has both genotypic and phenotypic information, for instance focusing on SNPs associations of tumor microenvironment (TME) metrics, but they did not focus on outcome.

In this work, Pagadala et al performed an impressive analysis using the TCGA as reference and the UK Biobank and immune checkpoint inhibitor datasets as validation. Importantly, they use the current literature (recent germline studies) as baseline to extract additional information.

Results are extremely intriguing as authors found germline-TIME association associated with cancer outcome and immunotherapy response. The findings expand significantly current knowledge in the field. In addition to the wide genomic analysis authors were also able to validate one of the target (CTSS) through functional experiments, which represents a good proof of principle.

Combining and analyzing these large datasets from a germline genetic point of view is extremely complex and authors should be commended for their effort. The data processing is meticulous and also include WES for QC control.

Overall methods are well described (although some clarifications are needed) and the manuscript is well written, also considering the complexity and the extent of the analysis.

Overall, this is an extremely important report that will spur novel investigations and clinical application of germline genetic in the cancer immunotherapy space.

There are however, in my opinion, some points that need to be addressed before publication.

- 1) An intrinsic limitation is represented by the fact that authors did not use rare variants, and again, this is understandable as the TCGA germline exome data suffer a considerable problem in term of batch effects, but such limitations might be discussed.
- 2) Methods are well described, yet not easy to follow as authors assess multiple datasets. For clarity, each paragraph should describe (for instance in the title) to which datasets does this specific method apply to.
- 3) It is difficult to follow which covariate adjustment have been performed for each analysis and this should be better explained as it is a critical component. It might be useful to include, in the summary figure describing the overall approach: 1) The exact cohort/dataset used, 2) the covariate adjustment. If it becomes difficult to include all these information in one figure a supplementary figure might be added. In the TCGA, adjusting for cancer-type is mandatory and authors should better declare how this was performed and in which analyses.
- 4) In general, quantification of TIME subpopulations could be performed by analyzing the proportion of different leukocyte populations (ie, through CIBERSORTX), or the estimate quantities (obviously in a relative and not absolute way), using ssGSEA for each subpopulation. In Sayaman et al, CIBERSORT traits were not heritable. CIBERSORT tends to give estimates equal to zero in a big number of samples for some subpopulations. CIBERSORTx (used by the authors) might have been more accurate. While authors mentioned that "Phenotypes with greater than 10% zero values were excluded", authors should define in the text more exactly which CIBERSORTx traits were included in the final analysis. Authors should provide a table with all the per sample values of the phenotypes analyzed (as for paragraph "TCGA Phenotype Data")
- 5) In the methods, authors should better indicate which phenotypes were recalculated and which ones were downloaded from Thorsson et al. The current sentence "The following phenotypes were extracted or generated from RNA-seq data" is not clear.
- 6) The method section "TCGA Phenotype Data", does not contain references for many phenotypes (ie, Danaher et al, which likely refers to Danaher et al, JITC, 2015). Please indicate the exact reference. In that paper (Danaher et al) authors defined a list of 60 marker genes whose expression levels measure 14 immune cell populations. Do the authors assess each marker independently or did they use these

markers for cell population estimations? I understood each markers was assessed independently but this needs to be specified.

7) TCGA genotype arrays were represented by an early version of Affymetrix arrays, which include a limited number of SNPs, the imputation and filtering is a particularly complex process and authors should be commended for having processed and performed the imputation. An important problem of TCGA germline data is that they are restricted access so, even if a group impute the missing SNPs, the imputed matrix cannot be shared as is subject to the same limitation of the mother file, which require dbGAP authorization. At the moment, the only resource of filtered and imputed genotype data is represented by the one by Sayaman et al, as it was linked to a TCGA Network Ancestry Paper. Authors are however encouraged (if they believe it is appropriate; this is not a mandatory request) to check with the TCGA whether there is any way to share this important resource of imputed data for other scientists.

8) The authors mapped SNPs to gene based on the position of the SNPs in respect to the gene which might not be accurate in absence of colocalization analysis. I'm not requesting to perform colocalization analysis, which would probably require another manuscript per se; still, in the text authors should perhaps label differently SNPs that are intronic or vs the ones that are in the protein coding part and can therefore attributed uniquely to a specific gene. This limitation should be discussed.

9) At page 39, among the Literature TIME-SNPs manuscripts used to compile the list of existing germline variants associations, authors listed the ms by Bedognetti et al which checked CXCR3 polymorphisms and CCR5 delta 32 deletion. Did the authors also imputed indels (such as CCR5 delta 32 deletion) or only SNPs?. This should be specified.

Reviewer #2:

Remarks to the Author:

Pegadala et al perform a large scale analysis of TCGA to examine genetic variants underlying tumor immune microenvironment (TIME). They identify many variants associated with TIME and then evaluate how these variants predict cancer susceptibility, survival and/or response to immunotherapy. Analyses of germline variation in relation to TIME could be used to (a) identify new genes/pathways that are important in immune response and (b) to predict individuals who might respond better/worse to checkpoint inhibitors. Thus, the work is potentially significant. However, the manuscript, in its current form, has significant limitations as below:

1. The authors talk about TIME but the vast majority of the signatures that they analyze (>95% of the signatures listed in Table S1) are single gene expression. Thus, most of the analyses boil done to simple eQTL analyses. While these may be of interest they should not be classified SNPs that clearly affect TIME. For example, a SNP may affect the expression of a gene that functions in an immune cell but that SNP may not drive an overall change in the pattern of cells and/or the cytokine patterns in IME. The authors should clarify that most of these results are eQTLs of genes that COULD affect TIME but do not necessarily do that.

2. The large number of SNPs may be misleading (esp at HLA where there is remarkable LD). The authors should perform LD clumping and/or some other method to describe the number of independent loci associated with immune signatures of interest. This should be done throughout the manuscript. For example Fig 3A shows a large number of SNPs that overlap between SNP-TIME associations. But it may be that all 37 SNPs associated with antigen presentation and survival are at HLA and represent one haplotype.

3. The analyses with survival are potentially of major importance. However, analysis of survival in TCGA is potentially subject to overfitting in the sense that the authors are identifying the SNP-immune signature associations and the SNP-survival associations in the same dataset and they are starting with immune signatures known to be associated with survival. Thus, the UKB associations with survival should be taken with much more

4. The authors use false discovery rates that vary from 0.05 to 0.5 and many other values in between.

(For example line 272 $FDR < 0.25$ and line 274 $FDR < 0.5$ line 330 $FDR < 0.2$, line 332 $FDR < 0.17$ and other instances in the paper). A clear and rigorous threshold for multiple hypothesis testing is critical. FDR of 0.2, 0.25 0.5 etc should not be reported as "significant" and should probably not even be included as a main result.

5. The manuscript reads as a laundry list of associations. If the goal is to understand biology of the immune response, the authors should seek to infer in a more rigorous and comprehensive way what the pathways and mechanisms that are suggested behind their associations.

6. The authors do a good job of reviewing prior literature and integrating prior analyses into theirs. However, Sayaman et al (citation #9) cites a Biorxiv manuscript. The main text cites a published paper that appears to be peer reviewed version of the manuscript. Presumably, the authors used the peer reviewed version for their SNP selection. If so, citation #9 should be the published version.

Reviewer #3:

Remarks to the Author:

The authors sought to identify germline variants associated with tumor microenvironment using 194 literature-curated TIME associations and 890 associations detected with 157 immune phenotype (IP) components found using genotypes from over 8,000 individuals in The Cancer Genome Atlas (TCGA). As a follow-up, the authors also investigated the identified associations with cancer risk, survival and immune response outcomes. Intensive data analyses of many different kinds were conducted. Mouse data on CTSS were presented. This work provided potentially useful information on the plausible effects of inherited genetics on tumor microenvironment and cancer outcomes. Some comments on several major analytic issues.

Identifying heritable characteristics of the tumor immune microenvironment (TIME)

1) The immune phenotype (IP) analysis using expression and RNASeq data as phenotypes was performed using 8000+TCGA samples, which are heterogeneous and consist of many different cancers. How did the analysis take between-cancer heterogeneity into account? Given different cancers have different underlying biology, some justification would be useful to show that the pooled SNP-TIME(IP) analyses are valid by combining data from many different cancers, and the results are robust across cancer types.

2) An immune phenotype (IP) was approximated based on tumor RNASeq data in two ways, 1) IP composite based on bulk RNASeq; 2) gene expression of single immune-related genes. As direct measurements of immune-related protein levels are often of more clinical interest, can you comment on the degree of consistency of IP-based and protein-based findings you would expect, and how the IP-based findings are likely to shed light on biomarker-based findings? Is it possible to validate some of the IP-based findings using some biomarkers, e.g. PD-L1 TPS?

3) It is interesting to estimate the heritability of IP separately 1) by whole-genome excluding HLA region and 2) by HLA locus.

a) How did you decide the cutoff of choosing heritability $> 5\%$? Any reference or justification? How are the results sensitive to different cutoffs?

b) Can you elaborate on what types of IPs are likely to have relatively high heritability and what are the major sources of missing heritability?

c) Is there any difference across different cancer types regarding the heritability? Given 8000 samples from different cancers were used in pooled heritability analysis and heritability is likely to vary across cancer types, how to justify the validity of the pooled heritability analysis and how to interpret the

pooled findings?

4) Among the detected significant associations, have you seen any enrichment for certain organs or systems? Any enrichment in biological function?

Identification and characterization of TIME-SNPs related to cancer outcomes

5) The authors evaluated the identified SNP-TIME associations with several cancer outcomes, including cancer risk, progression and response to immunotherapy. The risk factors, confounders, histology, and clinical characteristics (e.g. stage and subtypes), that affect disease risk, progression and response to immunotherapy are quite different and variable for different cancer types. It is not clear what covariates were controlled for in these analyses, and whether key clinical covariates were properly controlled for. Given the heterogeneity across different cancer types, it would be a bit worrisome to analyze many cancer-related outcomes in the pan-cancer setting if only age and sex were used as covariates. It would be useful if the authors can elaborate what covariates that were used in the analysis, and perform more thorough analysis of a few cancer types. For example, for progression analysis of lung cancer, one could consider including baseline risk factors (e.g. age, gender) and key confounders, e.g. smoking history for lung cancer risk; clinical stage, and treatment information for PFS and OS.

Variants underlying immune evasion are associated with cancer survival

6) As an example, majority of the lung cancer patients in TCGA were early-stage patients, thus the PFS needs to be defined clearly. This is because PFS is highly treatment dependent and should be defined clearly in different analyses, e.g. whether time 0 of PFS is defined as the time of surgery or any systemic treatment, especially in the response to ICB analyses.

7) "We also found the burden of immune checkpoint variants to be significantly associated with overall survival in lung adenocarcinoma (Figure 5E)." Once again, can you comment on what covariates, such as key clinical covariates, were controlled in this analysis? For example, the overall survival of early-stage lung cancer patients depends on many important clinical factors and the treatments received.

Immune Checkpoint Blockade (ICB) Response Analysis

8) The authors used melanoma patients as discovery while renal cancer and non-small cell lung cancer cohort as validation. This is problematic as these are very different types of solid tumors. It seems that only age and sex were controlled in the analysis.

9) The germline variant burden score was defined as the total number of variants each patient carried, which ignored the effect sizes. Is there any reason for this definition? Why not using PRS calculated using these variants that can incorporate effect sizes.

10) The author conducted experiment in the MC38 murine model. Given many analyses were performed, why did you select CTSS over other genes in generating the experimental data? Why not other genes, such as PSMD11?

Analysis inconsistency

11) The authors performed many analyses using data from many sources. I appreciate these efforts. In the meantime, some of statistical analyses seem to be inconsistent, and various ad hoc decisions were made. For example, results using a wide range of FDR cutoffs were reported for different analyses, ranging from $FDR < 0.05$, 0.2, 0.25 or 0.5. This gives the reader a feeling of fishing, e.g., if nothing was significant at the FDR 0.05 level, a more liberal FDR level was used in order to report "significant" findings. It would be useful if consistent statistical analyses and criteria can be used.

Reviewer #4:

Remarks to the Author:

Pagadala et al. present an interesting analysis of germline genetic variants associated with immune phenotypes, and the overlap of these associations with cancer risk and treatment outcomes in immunotherapy.

The authors assess TIME-associated germline variants curated from literature, and identify further SNPs in an analysis of the TCGA database. Out of this set of variants, approx. 20% are shown to be cancer-relevant, i.e. associated with cancer risk or ICB outcome. As an example, CTSS inhibition was shown to improve survival in a mouse model.

The paper is timely and relevant for the field, also because the application of germline genetics is still difficult in the field of cancer immunotherapies, where large cohorts for hypothesis-free analyses (e.g. GWAS) are not yet available / of sufficient size. Thus, starting with an immune-relevant subset of variants is a promising approach.

That said, there are a couple of concerns I'd like to address:

Major:

- The figures and overall presentation of the paper suggest that "all associations are created equal" (e.g. Figure 3A), although different FDR cutoffs for multiple testing correction were used, depending on the analysis and (likely) power considerations. I don't think that is entirely appropriate. While I wouldn't necessarily insist on Bonferroni corrections in a discovery setting where I'm interested in overlapping associations for multiple phenotypes, some choices appear overly lenient (FDR-corrected $p=0.25$ or even $p=0.5$ for ICB outcomes).

Minor:

- The authors use the term "IP component" to describe the set of variables they test for association with germline genetic variation. In the majority of cases, these components are just RNA expression levels of single genes. Wouldn't it be more straightforward to just refer to eQTL analyses instead, and to use the term IP components only for the more complex variables? For example, on page 7 the authors refer to "GWAS analysis for 17 IP components corresponding to genes in the HLA region of chromosome 6" -> "GWAS for HLA gene expression levels" or similar wouldn't require reading it several times.

- Figure 1A is very helpful to better understand the analysis strategy, but it is sometimes difficult to follow the logic in the text. For example, on page 7 the authors describe 825 associations with 75 IP components, but then write that those are based on a suggestive multiple testing threshold, with only a subset reaching genome-wide significance. Given that there is no scarcity of significant associations, it might be better to start with the genome-wide significant associations, and then expand.

- HLA genes eQTL analysis: HLA-DRB3,4,5 were included, but they are copy-number variable and in LD with specific HLA-DRB1 alleles. Did the authors check if expression estimates are credible (e.g.: there should be a significant number of individuals with no HLA-DRB3,4, or 5 expression).

- Figure 4C: p values missing

- Figure 5E: Are the p values for the pairwise comparisons switched? (low/medium and medium/high)

- I appreciate the additional functional analysis component of the paper, but the CTSS analyses appear a bit out of place. It's true that there's not much known about CTSS in solid tumors, but the results are also not surprising given what was shown previously e.g. for follicular lymphoma. Also, CTSS has been investigated in autoimmune settings, showing consequences on macrophage function and gene expression levels (Review by Brown et al., Respiratory Research, 2020). In solid tumors, CTSS inhibition was shown to impact TGF-beta-related pathways (Li et al., J Cancer, 2021), autophagy

(Fei et al., Front Oncol, 2020) and BRCA1 stability (Kim et al., Cell Death and Differentiaion, 2019). So there is a lot to consider when investigating CTSS on a functional level in the context of immune-checkpoint blockade. That said, the results related to combination therapy with anti-PD1 are interesting, and I'm aware that reviewers tend to ask for additional "functional insight" in manuscripts focused on statistical associations.

- About a third of the introduction is already a summary of the results. Maybe this part can be shortened.

- Abstract: Too many numbers and a bit confusing. E.g.: "we evaluated 194 literature-curated TIME associations and 890 associations detected with 157 immune phenotype (IP) components". Maybe just write "1084 associations of TIME SNPs with..." and explain later? Also, "IP components" should be clearly defined if used in the abstract.

Chris Hammer, June 2022

REVIEWER COMMENTS

Reviewer #1, expert in cancer immunogenetics (Remarks to the Author):

The extent and characterization of germline genetic contribution to cancer immune responsiveness remains largely uncharacterized. So far three large genome-wide analyses have been performed using the TCGA dataset which has both genotypic and phenotypic information, for instance focusing on SNPs associations of tumor microenvironment (TME) metrics, but they did not focus on outcome. In this work, Pagadala et al performed an impressive analysis using the TCGA as reference and the UK Biobank and immune checkpoint inhibitor datasets as validation. Importantly, they use the current literature (recent germline studies) as baseline to extract additional information. Results are extremely intriguing as authors found germline-TIME association associated with cancer outcome and immunotherapy response. The findings expand significantly current knowledge in the field. In addition to the wide genomic analysis authors were also able to validate one of the target (CTSS) through functional experiments, which represents a good proof of principle. Combining and analyzing these large datasets from a germline genetic point of view is extremely complex and authors should be commended for their effort. The data processing is meticulous and also include WES for QC control. Overall methods are well described (although some clarifications are needed) and the manuscript is well written, also considering the complexity and the extent of the analysis.

Overall, this is an extremely important report that will spur novel investigations and clinical application of germline genetic in the cancer immunotherapy space. There are however, in my opinion, some points that need to be addressed before publication.

We thank the reviewer for the accurate summary of our work, the recognition of the effort it entails and the very helpful suggestions.

1) An intrinsic limitation is represented by the fact that authors did not use rare variants, and again, this is understandable as the TCGA germline exome data suffer a considerable problem in term of batch effects, but such limitations **might be discussed**.

We now addressed the omission of rare variants in the discussion of our manuscript as follows:

“Our analysis had several limitations. We focused on common germline variation; however, rare germline variants have potential to modify the tumor immune microenvironment. In Sayaman et al., MMR rare variants were associated with higher lymphocyte infiltration and *BRCA1* mutations with IFN and MHC response modules¹. Exploration into rare variants in immune genes could reveal aspects of TIME but might also share mechanisms with increased infection rates or immunodeficiencies^{2,3}. These individuals may be affected by rare cancer types as observed in transplant and HIV-infected patients⁴”

2) Methods are well described, yet not easy to follow as authors assess multiple datasets. For clarity, each paragraph should describe (for instance in the title) to which datasets does this **specific method apply to**.

We have reorganized the methods and now describe the relevant dataset in each section. For example:

Heritability Analysis - TCGA GCTA Analysis

Heritability estimates were calculated with the genomic-relatedness-based restricted maximum-likelihood (GREML) approach implemented in GCTA (Genome-wide Complex Trait Analysis)^{5,6}. Genetic

relationship matrices (GRMs) which measure genetic similarity of unrelated individuals (GRM <0.05) were constructed for the autosomal and X chromosomes for the TCGA European ancestry samples. Benjamini-Hochberg false discovery rates (FDR) were calculated using statsmodels⁷. Immune traits were considered sufficiently heritable if the V(g)/V(p) value was > 0.05 using the full GRM.

Risk Analysis - UK Biobank

To assess cancer risk, we conducted PheWAS with cancer ICD10 codes in the UK Biobank. UK Biobank subjects were subsetted into separate ethnic-racial groups following continental ancestry prior to analysis. The sub-setting was performed to generate homogenous groups and reduce potential admixture bias in the genetic analyses.

3) It is difficult to follow which covariate adjustment have been performed for each analysis and this should be better explained as it is a critical component. It might be useful to include, in the summary figure describing the overall approach: 1) The exact cohort/dataset used, 2) the covariate adjustment. If it becomes difficult to include all these information in one figure a supplementary figure might be added. In the TCGA, **adjusting for cancer-type is mandatory and authors should better declare how this was performed and in which analyses.**

We have updated our summary figure to include the covariate adjustment for the main analysis (**Figure R1**) and added supplementary tables detailing covariates used for tumor type-specific analyses (**Supplementary Table 10**).

Figure R1: The updated Figure 1 now includes covariate information.

To control type I error that can arise from non-normality of residuals in linear regression analysis, we inverse rank-normalize IP components within each cancer type. This also results in the values being identically distributed for each TCGA cancer type. An example is shown below for the FGR gene IP component (**Figure R2**).

Figure R2: Boxplot of rank-normalized FGR expression by TCGA cancer type.

To verify that this procedure controls for tumor type effectively, we evaluated the 1084 TIME SNPs assessed in our study for tumor type association, and found only 1 (rs146336885) was significantly associated with tumor type ($p < 5e-08$) (**Table R1, Supplementary Table 6 in manuscript**).

We have sought to further clarify this in the text as follows:

“Immune gene expression was inverse-rank normalized within tumor type, such that tumor-type specific differences were removed.”

“Finally, to confirm TIME eQTLs were not cancer-type specific, we conducted associations with tumor type. Of our 890 TIME eQTLs, only rs146336885 was associated with tumor type (**Supplementary Table 6, Figure S3G**).”

Table R1. SNPs significantly associated with tumor type.

SNP	REF	ALT	OR	P	Cancer Type
rs146336885	G	A	5.91539	1.08E-18	TGCT
rs146336885	G	A	2.08505	1.10E-19	KIRC
rs146336885	G	A	0.204725	4.75E-10	PCPG
rs146336885	G	A	4.7861	9.82E-09	CHOL
rs146336885	G	A	0.307229	3.35E-20	HNSC
rs146336885	G	A	1.68105	1.35E-10	OV

In response to R2 we have now also added additional covariates to the survival analysis. We sought to clarify which covariates were used for what analysis in the methods, and added **Supplementary Table 10** specifically to describe tumor-type specific covariates used for survival analysis.

4) In general, quantification of TIME subpopulations could be performed by analyzing the proportion of different leukocyte populations (ie, through CIBERSORTX), or the estimate quantities (obviously in a relative and not absolute way), using ssGSEA for each subpopulation. In Sayaman et al, CIBERSORT traits were not heritable. CIBERSORT tends to give estimates equal to zero in a big number of samples for some subpopulations. CIBERSORTx (used by the authors) might have been more accurate. **While authors mentioned that “Phenotypes with greater than 10% zero values were excluded” , authors should define in the text more exactly which CIBERSORTx traits were included in the final analysis.** Authors should provide a table with all the per sample values of the phenotypes analyzed (as for paragraph “TCGA Phenotype Data”)

Although we used CIBERSORTx, the majority of infiltrates still had zero values in >10% of samples, and similar to the report by Sayaman, the remainder did not show evidence of heritability. We have now highlighted in Supplementary Table 1 the IP components which were filtered out due to having greater than 10% zero values. These include the following CIBERSORTx immune infiltrate estimates:

B cells naive
B cells memory
Plasma cells
T cells CD4 naive
T cells CD4 memory resting
T cells CD4 memory activated
T cells regulatory (Tregs)
T cells gamma delta
NK.cells.resting
NK cells activated
Monocytes
Macrophages M0
Macrophages M1
Dendritic cells resting
Dendritic cells activated
Mast cells resting
Mast cells activated
Eosinophils
Neutrophils

The remaining infiltrate IP components were not designated as heritable by our thresholds and thus not analyzed downstream.

T cells CD8
T cells follicular helper
Macrophages M2

However, through inclusion of literature SNPs, we did test a SNP listed by Shahamatdar as associated with T follicular helper cells which was ultimately implicated in immunotherapy response. As suggested by the reviewer,

we have included **Supplementary Table 2** with all cancer type rank-normalized IP component values provided for the TCGA.

To better clarify that the majority of IP components analyzed ended up being simple eQTLs, we included the following:

“No composite phenotypes passed heritability thresholds and thus remaining associations were with gene expression and will be referred to as TIME eQTLs.”

We opted to retain the composite phenotypes in the initial screen despite the lack of any meeting statistical inclusion criteria as it may be instructive for future efforts that wish to look for SNP associations with such.

5) In the methods, authors should better indicate which phenotypes were recalculated and which ones were downloaded from Thorsson et al. The current sentence “The following phenotypes were extracted or generated from RNA-seq data” is not clear.

To clarify which phenotypes were extracted or generated, we have now added a column to Supplementary Table 1 which specifies: 1) Generated from Pancan Atlas RNA-seq or 2) Extracted from Thorsson *et al.*

6) The method section “TCGA Phenotype Data”, **does not contain references for many phenotypes (ie, Danaher et al, which likely refers to Danaher et al, JITC, 2015)**. Please indicate the exact reference. In that paper (Danaher et al) authors defined a list of 60 marker genes whose expression levels measure 14 immune cell populations. Do the authors assess each marker independently or did they use these markers for cell population estimations? I understood each marker was assessed independently but this needs to be specified.

We have now updated the TCGA Phenotype Data section to include references for all phenotypes, including Danaher et al, Thorsson et al and Newman et al.

- **Immunomodulators:** 436 genes used to define immune states from Thorsson et al⁸.
- **Immune checkpoint molecules:** 78 immune checkpoint stimulatory and inhibitory molecules from Thorsson et al⁸.
- **Antigen Presentation:** 231 antigen presentation genes from Gene Ontology [GO_REF:0000022]
- **Immune cell markers:** 60 immune cell type markers from Danaher et al⁹.
- **IFN- γ :** IFN- γ genes retrieved from Biocarta [Systematic Name: M18933]
- **TGF- β :** TGF- β genes retrieved from Biocarta [Systematic Name: M22085]
- **Immune states:** Individual level scores for 6 immune states [wound healing, IFN- γ dominant, inflammatory, lymphocyte depleted, immunologically quiet, and TGF- β dominant] from Thorsson et al⁸.
- **Immune infiltration levels:** 22 relative immune infiltration estimates from CIBERSORTx¹⁰ using LM22 signature matrix.

We have also clarified in the manuscript that each gene was assessed independently.

“To describe the TIME, we collected a comprehensive set of immune phenotype (“IP”) components comprising composite measures derived from bulk gene expression and expression levels of individual immune-related genes (**Figure 1B**). Composite phenotypes included infiltrating immune cell levels calculated using CIBERSORTx (immune infiltrates) and 6 immune subtype scores from a pan-cancer TCGA analysis by Thorsson *et al.* (landscape components). Immunomodulators were collected from Thorsson *et al.*, where weighted gene correlation network analysis was used as an unbiased systematic approach to identify gene sets relevant to the TIME. We included genes from these sets along with immune checkpoint genes, cell type markers, antigen presentation genes, TGF- β pathway genes, and IFN- γ genes as these have been implicated as important modifiers of the TIME. After removing IP components with high numbers of zero values to reduce spurious associations, we retained 724 immune-related genes and 9 composite phenotypes (733 IP components total) measured across 30 cancer types (**Supplementary Table 1-2, Figure S1, Table S1**). Each IP component (gene expression level or composite phenotype) was analyzed independently.”

7) TCGA genotype arrays were represented by an early version of Affymetrix arrays, which include a limited number of SNPs, the imputation and filtering is a particularly complex process and authors should be commended for having processed and performed the imputation. An important problem of TCGA germline data is that they are restricted access so, even if a group impute the missing SNPs, the imputed matrix cannot be shared as is subject to the same limitation of the mother file, which require dbGAP authorization. At the moment, the only resource of filtered and imputed genotype data is represented by the one by Sayaman et al, as it was linked to a TCGA Network Ancestry Paper. Authors are however encouraged (if they believe it is appropriate; this is not a mandatory request) to check with the TCGA whether there is any way to share this important resource of imputed data for other scientists.

We have contacted the TCGA and are negotiating to publish imputed genotypes under a publication resource page. This should make it possible for individuals with approved controlled access to download the genotype matrix used for our study. Currently, the text still reads that data can be requested directly from the authors (which will always be true). We are optimistic we will be able to add a URL for a publication resource page in the coming weeks.

8) The authors mapped SNPs to gene based on the position of the SNPs in respect to the gene which might not be accurate in absence of colocalization analysis. I’m not requesting to perform colocalization analysis, which would probably require another manuscript per se; still, in the text authors should perhaps label differently SNPs that are intronic or vs the ones that are in the protein coding part and can therefore attributed uniquely to a specific gene. This limitation should be discussed.

We have included a description of the number of variants affecting protein-coding regions.

“Eight cancer relevant TIME-SNPs (1.6%) affected protein-coding regions (**Figure S7B**). In the case of *HLA-A*, *HLA-C*, *FPR1*, *CTSS*, *TAP2*, missense variants in coding regions were associated with expression differences. In addition, missense variants in *PALB2*, *NOTCH4* and *GBP3* were associated with expression differences in *DCTN5*, MHC Class II and *CCBL2*, respectively (**Figure S7C**).”

We added 4 columns to **Supplementary Table 4** to highlight SNPs which are protein-coding or affecting NMD, splice and/or transcription factor binding sites. We also have included VEP annotations for genes.

We have added the following to the discussion:

“SNP to gene linkages were assumed based on SNP association with gene expression. However, it is possible that some SNPs may be incorrectly linked to target genes or may affect the expression of multiple genes.”

9) At page 39, among the Literature TIME-SNPs manuscripts used to compile the list of existing germline variants associations, authors listed the ms by Bedognetti et al which checked CXCR3 polymorphisms and CCR5 delta 32 deletion. Did the authors also imputed indels (such as CCR5 delta 32 deletion) or only SNPs?. This should be specified.

We restricted our analysis to only SNPs, thus CCR5 delta 32 deletion was not included in our analyses. As clarification, we also included a statement in methods under “TCGA Genotype Processing”:

“Only single nucleotide polymorphisms (SNPs) were analyzed.”

Reviewer #2, expert in human population genetics and epidemiology (Remarks to the Author):

Pegadala et al perform a large scale analysis of TCGA to examine genetic variants underlying tumor immune microenvironment (TIME). They identify many variants associated with TIME and then evaluate how these variants predict cancer susceptibility, survival and/or response to immunotherapy. Analyses of germline variation in relation to TIME could be used to (a) identify new genes/pathways that are important in immune response and (b) to predict individuals who might respond better/worse to checkpoint inhibitors. Thus, the work is potentially significant. However, the manuscript, in its current form, has significant limitations as below:

1. The authors talk about TIME but the vast majority of the signatures that they analyze (>95% of the signatures listed in Table S1) are single gene expression. Thus, most of the analyses boil down to simple eQTL analyses. While these may be of interest they should not be classified SNPs that clearly affect TIME. For example, a SNP may affect the expression of a gene that functions in an immune cell but that SNP may not drive an overall change in the pattern of cells and/or the cytokine patterns in TIME. The authors should clarify that most of these results are eQTLs of genes that COULD affect TIME but do not necessarily do that.

We thank the reviewer for highlighting this important point. Although we included more complex phenotypes at the early stages, these did not pass QC and SNP heritability checkpoints. Thus, other than the literature SNPs, the majority of SNPs we studied are indeed immune eQTLs. We have clarified this in the manuscript as follows:

“No composite phenotypes passed heritability thresholds and thus remaining associations were with gene expression and will be referred to as TIME eQTLs.”

We agree that many of these SNPs will be simple immune eQTLs with no relevance to cancer per se. That is why we performed a second analysis evaluating TIME eQTL SNPs with respect to cancer risk, progression and therapy response. Focusing on SNPs that both associate with immune gene expression in the TIME and cancer risk, progression or response to therapy is similar in spirit to a TWAS analysis intersecting eQTLs with GWAS hits. The SNPs associated with both expression and cancer outcomes are much more likely to indicate causal loci. We have sought to clarify these ideas in the manuscript as follows:

“An association with gene expression in the TIME does not necessarily mean that the eQTL will impact cancer outcomes.”

“Colocalization of gene expression and GWAS signals can point to putative causal disease-related genes that in the setting of ICB response might suggest candidate targets to stimulate more effective anti-tumor immunity.”

Clarification

2. The large number of SNPs may be misleading (esp at HLA where there is remarkable LD). The authors should perform LD clumping and/or some other method to describe the number of independent loci associated with immune signatures of interest. This should be done throughout the manuscript. For example Fig 3A shows a large number of SNPs that overlap between SNP-TIME associations. But it may be that all 37 SNPs associated with antigen presentation and survival are at HLA and represent one haplotype.

In order to ensure our results represent independent associations, we performed LD clumping using the primary suggestive threshold corrected for the number of phenotypes tested ($1 \times 10^{-5}/140$) using a kb threshold of 500, and an R^2 threshold of 0.5.

Additionally, since the HLA region is highly polymorphic, we conducted a conditional analysis where we ran stepwise conditional analysis to retain only independent HLA associations. In this approach the most significant initial associations detected with HLA region phenotypes by standard GWAS analysis are incorporated as covariates in subsequent rounds of association analysis¹¹. Specifically, we re-ran the analysis with chromosome 6 variants including the most significant SNP (lowest p-value in the previous round) as a covariate. Analysis was conducted until no SNPs with Bonferroni-corrected p-value $< (1 \times 10^{-5}/17)$ remained. This procedure greatly reduced the number of significant SNPs from 90,549 initial to 65 final independent associations. Results are also shown in **Figure 2B**.

We have updated the manuscript as follows:

“To remove HLA region associations solely attributable to LD structure^{11,12}, we conducted conditional GWAS analysis for seventeen genes in the HLA region of chromosome 6. Alignment to a general HLA gene reference can introduce error into expression level estimates due to the highly polymorphic nature of these genes. We therefore also revisited SNP associations with gene expression estimates derived from allele-specific RNA alignments¹³ (**Methods**) and performed GWAS analysis using allele specific expression. In total, we identified 65 TIME eQTLs in the HLA region (**Figure 2B**).”

3. The analyses with survival are potentially of major importance. However, analysis of survival in TCGA is potentially subject to overfitting in the sense that the authors are identifying the SNP-immune signature associations and the SNP-survival associations in the same dataset and they are starting with immune signatures known to be associated with survival. Thus, the UKB associations with survival should be taken with much more

The reviewer raises an important point. Although we did not select immune genes and composite phenotypes based on any survival association, we cannot completely rule out that they were included in the prior studies we curated because of such associations. The genes and composite phenotypes used were collected from the following sources:

- **Immunomodulators:** 436 genes used to define immune states from Thorsson et al⁸.
- **Immune checkpoint molecules:** 78 immune checkpoint stimulatory and inhibitory molecules from Thorsson et al⁸.
- **Antigen Presentation:** 231 antigen presentation genes from Gene Ontology [GO_REF:0000022]
- **Immune cell markers:** 60 immune cell type markers from Danaher et al⁹.
- **IFN- γ :** IFN- γ genes retrieved from Biocarta [Systematic Name: M18933]
- **TGF- β :** TGF- β genes retrieved from Biocarta [Systematic Name: M22085]
- **Immune states:** Individual level scores for 6 immune states [wound healing, IFN- γ dominant, inflammatory, lymphocyte depleted, immunologically quiet, and TGF- β dominant] from Thorsson et al⁸.

- **Immune infiltration levels:** 22 relative immune infiltration estimates from CIBERSORTx¹⁰ using LM22 signature matrix.

Unfortunately, the UK Biobank does not provide survival information. There are actually not that many cohorts that provide both genome-wide genotype data and survival outcomes. Fortunately, we were able to obtain data from the Sherlock lung cancer study at NCI, including genotypes, clinical covariates and survival information for 166 never-smokers with lung adenocarcinoma. We fed the genotype data into the polygenic survival score (PSS) we trained on TCGA lung adenocarcinoma (LUAD) and found that this score was able to separate both the held out 30% of samples from TCGA LUAD and the independent Sherlock cohort samples according to survival outcomes (**Figure 4C-E**). We note that the PSS had a higher hazard ratio in the Sherlock validation cohort (**Figure R3**). Upon close inspection of the eQTLs included in the PSS, we noted the presence of folate metabolism-related genes DHFR, and GGH. To ensure our validation wasn't solely due to pharmacogenomic effects on anti-folate treatments commonly used in cancer therapy, but for which we did not have treatment information, we revisited our PSS excluding these two genes and found it remained significant (**Figure S5**). This supports that the survival associations based on TCGA are likely to generalize to other cohorts and not simply the result of overfitting the data.

Figure R3: TIME eQTLs associated with survival implicate immune evasion. (C) Overall survival Kaplan-Meier curve based on LUAD PSS in TCGA LUAD. **(D)** Overall survival Kaplan-Meier curve based on LUAD PSS in SHERLOCK. **(E)** Cox Proportional Hazards for LUAD PSS in TCGA LUAD and SHERLOCK.

4. The authors use false discovery rates that vary from 0.05 to 0.5 and many other values in between. (For example line 272 FDR<0.25 and line 274 FDR<0.5 line 330 FDR<0.2, line 332 FDR<0.17 and other instances in the paper). A clear and rigorous threshold for multiple hypothesis testing is critical. FDR of 0.2, 0.25 0.5 etc should not be reported as "significant" and should probably not even be included as a main result.

We have now simplified our analysis framework to use common criteria for all analyses. We have updated the language to ensure that individual SNP associations are only considered significant if they pass an FDR < 0.05. Supplementary tables include all associations annotated according to FDR to allow readers to explore other thresholds for individual SNPs. To avoid using arbitrary FDR cutoffs for burden score construction while also incorporating SNP effect sizes as suggested by Reviewer 3, we also updated our polygenic risk analysis. We now use a recent approach that performs shrinkage via Lasso to force coefficients for non-informative SNPs to 0 followed by a machine learning-based PRS capable of capturing non-linear effects (Elgart M, Lyons G, Romero-Brufau S, Kurniansyah N, Brody JA, Guo X, Lin HJ, Raffield L, Gao Y, Chen H, de Vries P. Non-linear

machine learning models incorporating SNPs and PRS improve polygenic prediction in diverse human populations. *Communications biology*. 2022 Aug 22;5(1):1-2.).

5. The manuscript reads as a laundry list of associations. If the goal is to understand biology of the immune response, the authors should seek to infer in a more rigorous and comprehensive way what the pathways and mechanisms that are suggested behind their associations.

Figure R4: Characterization of genes implicated by PICS model TIME eQTLs. A map of TIME eQTL biological functions, immune functions and cancer associations for 15 genes implicated as modifiers of immune checkpoint blockade response. Innate immune function indicates that TIME eQTLs are also DICE eQTLs for macrophages, monocytes or dendritic cells. Adaptive immune function indicates that TIME eQTLs are also DICE eQTLs for CD8+ T cells, CD4+ T cells or B cells. Risk indicates whether a gene was also implicated in PRS models. Survival indicates whether a gene was also implicated in PSS models. Asterisks (*) indicates that a small molecule inhibitor has been reported for a gene.

We have now summarized genes implicated by our ICB-associated TIME eQTLs with Figure 7 (**Figure R4 above**). This figure aims to summarize biological function, immune function (through analysis of cell-type associated effects on gene expression) and cancer relevance through risk and survival analysis for each gene. In this figure, we aim to highlight immune processes that contribute to ICB response in the context of their function and relevance to other aspects of anti-tumor immunity. The immune cell type eQTLs also suggest which genes may be acting more through innate (macrophages, dendritic cells) or adaptive (CD4+, CD8+ T cells and B cells) mechanisms.

This is described in the revised manuscript as follows:

“Re-visiting the 15 genes implicated by the PICS model (**Figure 7**), we sought to gain more perspective on the aspects of immunity influential for immunotherapy response. Many of these genes also had risk or survival associated eQTLs and were modifiers of gene expression in various immune cell types. Peptide processing appeared to be a major factor contributing to ICB responses, Peptidases involved in both class I (*ERAP1*, *ERAP2*) and class II (*CTSS*) peptide processing appeared to be a shared component between ICB response and risk. In contrast, aspects relating to cytolytic activity (*CTSW*), pathogen responses (*FPR1*, *C3A1* and *LYZ*) and single stranded DNA responses (*TREX1*) shared more in common between ICB response and progression. In contrast, eQTLs involving intracellular trafficking proteins *DCTN5* and *DYNLT1* appeared to uniquely affect ICB response. Interestingly, eQTLs for *DCTN5* showed immune cell type specific effects, whereas those for *DYNLT1* did not. These proteins mediate vesicle and organelle trafficking that may have different implications in different cell types. For example, in T cells they may play a role in immune synapse formation and energetics by transporting mitochondria to the membrane¹⁴. Interestingly, another vesicle trafficking gene, *VAMP3* was implicated in progression. Altogether, our analyses reveal a subset of TIME eQTLs that highlight key aspects of immune function with implications for cancer risk, progression and immunotherapy response”

6. The authors do a good job of reviewing prior literature and integrating prior analyses into theirs. However, Sayaman et al (citation #9) cites a Biorxiv manuscript. The main text cites a published paper that appears to be peer reviewed version of the manuscript. Presumably, the authors used the peer reviewed version for their SNP selection. If so, citation #9 should be the published version.

We thank the reviewer for catching the outdated Sayaman et al citation. We now cite the published manuscript.

Reviewer #3, expert in statistics for genetics and immunology (Remarks to the Author):

The authors sought to identify germline variants associated with tumor microenvironment using 194 literature-curated TIME associations and 890 associations detected with 157 immune phenotype (IP) components found using genotypes from over 8,000 individuals in The Cancer Genome Atlas (TCGA). As a follow-up, the authors also investigated the identified associations with cancer risk, survival and immune response outcomes. Intensive data analyses of many different kinds were conducted. Mouse data on CTSS were presented. This work provided potentially useful information on the plausible effects of inherited genetics on tumor microenvironment and cancer outcomes. Some comments on several major analytic issues.

Identifying heritable characteristics of the tumor immune microenvironment (TIME)

1) The immune phenotype (IP) analysis using expression and RNASeq data as phenotypes was performed using 8000+TCGA samples, which are heterogeneous and consist of many different cancers. How did the analysis take between-cancer heterogeneity into account? Given different cancers have different underlying biology, some justification would be useful to show that the pooled SNP-TIME(IP) analyses are valid by combining data from many different cancers, and the results are robust across cancer types.

As shown above in **Figure S1**, we rank-normalized phenotypes by cancer-type to control for between-cancer heterogeneity. This forced all phenotypes to have the same distribution within each tumor type, which should emphasize associations that are shared across cancer types at the expense of missing cancer-type specific effects. To confirm this, we ran associations with cancer type and found that only 1 variant (rs146336885) had any cancer-type associations. This is further supported by examining the cancer-type specific effect sizes (beta values) of TIME eQTLs with a heatmap, which show consistent effect direction across tumor types, with very few exceptions (**Figure R5, Figure S3G in revised manuscript**). These exceptions tend to occur in tumor types with smaller sample sizes such as cholangiocarcinoma (CHOL) and uterine carcinosarcoma (UCS). While these merit further investigation, it will likely require large numbers of tumors and some of these tumor types have very small sample sizes (< 60 tumors).

Figure R5: Clustermap of cancer type specific beta values for TIME eQTLs identified through pan-cancer analysis.

2) An immune phenotype (IP) was approximated based on tumor RNASeq data in two ways, 1) IP composite based on bulk RNASeq; 2) gene expression of single immune-related genes. As direct measurements of immune-related protein levels are often of more clinical interest, can you comment on the degree of consistency of IP-based and protein-based findings you would expect, and how the IP-based findings are likely to shed light on biomarker-based findings? Is it possible to validate some of the IP-based findings using some biomarkers, e.g. PD-L1 TPS?

We thank the reviewer for this comment. While protein levels are often more useful clinically, we did not have protein measurements available and it was not always clear in what cellular context the proteins would be most clinically informative. Our data suggest that for many of the implicated IP-based biomarkers, the expression may be contributing through specific immune cell populations, however the resolution to fully determine these populations is not attainable by either CibersortX or cross-referencing with the DICE database. We do note that *CTSS* has both high RNA levels and high protein levels in the Human Protein Atlas (**Figure R6**). *FPR1* was another gene with high RNA expression and at least medium/low protein expression. This might suggest that these genes will translate to clinically useful protein biomarkers in follow on studies.

Figure R6. Agreement between RNA levels and Protein levels in the human protein atlas for key TIME eQTLs across immune tissues.

3) It is interesting to estimate the heritability of IP separately 1) by whole-genome excluding HLA region and 2) by HLA locus.

a) How did you decide the cutoff of choosing heritability $> 5\%$? Any reference or justification? How are the results sensitive to different cutoffs?

The SNP-heritability cutoff requiring at least 5% of variance in IP-biomarker measurement to be explained by SNPs was based on Choi SW, Mak TS, O'Reilly PF. Tutorial: a guide to performing polygenic risk score analyses. Nature protocols. 2020 Sep;15(9):2759-72. They suggest that this is the minimal effect size for including SNPs in polygenic risk analyses such as those we perform later in our study.

If we raise the SNP-heritability threshold to 10%, 61 IP components still pass compared to 235 at the lower heritability threshold of 5%. Of the 15 genes implicated by risk, survival and ICB analysis, 7 had SNP-heritability $> 10\%$ suggesting multiple eQTLs useful in polygenic score modeling fall into the lower SNP-heritability ranges.

This is potentially consistent with a reported lack of agreement between strong eQTLs and causal GWAS loci. Mostafavi *et al* report that genes near GWAS hits are enriched in numerous functional annotations, are under strong selective constraint and have a complex regulatory landscape across different tissue/cell types, while genes near eQTLs are usually depleted of most functional annotations, show relaxed constraint, and have simpler regulatory landscapes (Mostafavi H, Spence JP, Naqvi S, Pritchard JK. Limited overlap of eQTLs and GWAS hits due to systematic differences in discovery. bioRxiv. 2022 Jan 1.). This may suggest that weaker eQTLs are more likely to contain bona-fide disease-associated loci, although it is not clear whether tumor-immune interactions are a major source of purifying selection that would significantly constrain immune eQTLs.

b) Can you elaborate on what types of IPs are likely to have relatively high heritability and what are the major sources of missing heritability?

Consistent with previous reports¹⁵ our data implicate ERAP1 and ERAP2 genes as having a very high proportion of gene expression attributable to polymorphisms. Other highly polymorphic regions such as HLA also show high Vg/Vp estimates with GCTA. Highly polymorphic regions tend to have inflated estimates of SNP-heritability¹⁶. This may contribute in part to the much higher estimates observed for the HLA locus. We note that gene expression measurements were taken from bulk tumor data which represents a heterogeneous mixture of cells, and eQTLs can be cell type specific. This could be one factor diluting the link between SNPs and gene expression levels. Furthermore, in the setting of cancer, dramatic remodeling of the DNA in tumor cells can alter the relationship between SNPs and gene expression levels (copy number changes, epigenetic silencing, etc). For example, in a 2013 study by Li et al, *cis*-acting eQTLs accounted for 1.2% of the total variation of tumor gene expression, while somatic copy-number alteration and CpG methylation accounted for 7.3% and 3.3%, respectively (Li Q, Seo JH, Stranger B, McKenna A, Pe'er I, LaFramboise T, Brown M, Tyekucheva S, Freedman ML. Integrative eQTL-based analyses reveal the biology of breast cancer risk loci. Cell. 2013 Jan 31;152(3):633-41.).

c) Is there any difference across different cancer types regarding the heritability? Given 8000 samples from different cancers were used in pooled heritability analysis and heritability is likely to vary across cancer types, how to justify the validity of the pooled heritability analysis and how to interpret the pooled findings?

Our strategy for detecting associations using within-tumor type inverse rank normalized phenotypes prioritized associations that broadly apply across many tumor types, both in terms of SNP-heritability estimation and detection of associations. Perhaps not surprisingly, a large number of the detected associations map to immune cell type specific eQTLs which are more likely to generalize across tissue sites. Further stratifying analysis of SNP-heritability of TIME characteristics by cancer type would be potentially very interesting but would require larger sample sizes for individual cancer types than are available through TCGA. Small sample sizes will result in unacceptably high standard error; for unrelated individuals and common SNPs, GCTA GREML SNP-heritability estimation requires at least 3160 unrelated samples to get the SE down to 0.1¹⁷.

We have updated the manuscript text to clarify how our approach might bias us to identify more general and immune-cell type specific factors and added the possibility of investigating tumor-type specific effects as a future direction in the discussion.

“Furthermore, our approach prioritized pan-cancer associations, which has the potential advantage of revealing more generalizable associations at the cost of missing cancer-specific effects.”

4) Among the detected significant associations, have you seen any enrichment for certain organs or systems? Any enrichment in biological function?

TIME SNPs implicated a number of genes involved in antigen presentation and innate immune responses. Several genes were involved in vesicular trafficking, antigen processing, cytolytic activity of T cells and responses to immunogenic stimuli such as pathogens or single stranded DNA. We have now tried to focus more on biological interpretation through our final section “” and the associated **Figure 7** (also presented here as **Figure R4**).

5) The authors evaluated the identified SNP-TIME associations with several cancer outcomes, including cancer risk, progression and response to immunotherapy. The risk factors, confounders, histology, and clinical characteristics (e.g, stage and subtypes), that affect disease risk, progression and response to immunotherapy are quite different and variable for different cancer types. **It is not clear what covariates were controlled for in these analyses, and whether key clinical covariates were properly controlled for {Table of covariates}**. Given the heterogeneity across different cancer types, it would be a bit worrisome to analyze many cancer-related outcomes in the pan-cancer setting if only age and sex were used as covariates. It would be useful if the authors can elaborate what covariates that were used in the analysis, and perform **more thorough analysis of a few cancer types**. For example, for progression analysis of lung cancer, one could consider including baseline risk factors (e.g. age, gender) and key confounders, e.g. smoking history for lung cancer risk; clinical stage, and treatment information for PFS and OS.

We agree with the reviewer and emphasize that we have not performed any pan-cancer analyses for cancer related outcomes. In the revised manuscript, we now provide more detail about the covariates used in each analysis. Specifically, we revised overview Figure S1. Furthermore, as suggested, we added additional cancer-type specific covariates for survival analyses to make the analysis more thorough. These are now detailed in **Supplementary Table 10**.

6) As an example, majority of the lung cancer patients in TCGA were early-stage patients, thus the PFS needs to be defined clearly. This is because PFS is highly treatment dependent and should be defined clearly in different analyses, e.g, whether time 0 of PFS is defined as the time of surgery or any systemic treatment, especially in the response to ICB analyses.

For TCGA survival analyses, we used the cleaned progression free and overall survival data published by Liu et al¹⁸. Although treatment information is very limited in TCGA, we did include tumor stage as a covariate.

7) “We also found the burden of immune checkpoint variants to be significantly associated with overall survival in lung adenocarcinoma (Figure 5E).” Once again, can you comment on what covariates, such as key clinical covariates, were controlled in this analysis? For example, the overall survival of early-stage lung cancer patients depends on many important clinical factors and the treatments received.

We evaluated polygenic scores in the context of Cox Proportional Hazards models where we included tumor-type specific covariates, now detailed in Supplementary Table 10. We have now included important covariates such as history of radiation therapy, targeted molecular therapy, pathologic stage and smoking status to LUAD analyses.

8) The authors used melanoma patients as discovery while renal cancer and non-small cell lung cancer cohort as validation. This is problematic as these are very different types of solid tumors. It seems that only age and sex were controlled in the analysis.

We thank the reviewer for raising this point. While the independent validation is done in different solid tumor types, they are all tumor types that respond to immune checkpoint blockade which points to shared potential for T cells to mount an antigen-directed response. Additionally, the TIME eQTLs we identified are not cancer-type specific and are enriched for immune cell-type specific eQTLs. Thus, we believe that we are identifying eQTLs that modify aspects of the shared tumor immune response that are more likely to generalize. We have added discussion to indicate this and clarify that our approach may miss aspects that are tumor type specific.

“Although tumor-immune interactions vary across tissue sites and tumor characteristics, our study design emphasized tumor-general effects, which may explain the generalization of the PICS across ICB cohorts with distinct tumor types.”

We have also clarified in the manuscript that we controlled for age, sex, PC1-10 in the discovery analysis of ICB-treated melanoma cohorts. We note that we are limited to the covariates reported by each of the ICB studies.

9) The germline variant burden score was defined as the total number of variants each patient carried, which ignored the effect sizes. Is there any reason for this definition? Why not using PRS calculated using these variants that can incorporate effect sizes.

We have now updated our polygenic score construction approach to incorporate effect sizes. To move away from using arbitrary FDR thresholds for determining which SNPs to include in each polygenic score, we adopted a recent state-of-the-art method that performs SNP selection using Lasso to remove uninformative or redundant SNPs, then trains a machine learning-based PRS capable of capturing non-linear relationships among SNPs. This approach was shown to outperform linear models such PRSice and LDpred on a number of complex phenotypes (Elgart M, Lyons G, Romero-Brufau S, Kurniansyah N, Brody JA, Guo X, Lin HJ, Raffield L, Gao Y, Chen H, de Vries P. Non-linear machine learning models incorporating SNPs and PRS improve polygenic prediction in diverse human populations. *Communications biology*. 2022 Aug 22;5(1):1-2.). To improve consistency of analysis and reporting, we use this approach for all polygenic score analyses in the revised manuscript (**Figure R7, Figure 5D in revised manuscript**).

Figure R7 ROC-AUC Curve Analysis for PICS trained on the discovery melanoma cohort with XGBoost based model and tested on Miao et al and Rizvi et al.

10) The author conducted experiment in the MC38 murine model. Given many analyses were performed, why did you select CTSS over other genes in generating the experimental data? Why not other genes, such as PSMD11?

We thank the reviewer for this question. There were a number of factors that went into deciding which gene to validate. An ideal gene for validation studies would be one where the linked SNP's association with ICB response was shared across that majority if not all of the immunotherapy cohorts, where the allele associated with higher expression was also the one associated with worse outcome (it is easier to suppress activity than restore it once lost) and where a small molecule inhibitor was already available. Furthermore, we wanted to test a gene that was not already well established as a modulator of immune response. We summarize these criteria for the ICB PRS genes in **Table R2 (Supplementary Table 14 in revised manuscript)**.

Table R2. Criteria for selecting a target for validation among genes implicated by the polygenic ICB score.

Gene	Best SNP agreement across human cohorts	Potential for inhibition (+ indicates inhibition may improve response)	Small molecular inhibitor available	Already a high priority immune target in solid tumors
CTSS	-----	+---+	Yes	No
TREX1	----+-	+++--	First in 2021	Yes
DHFR	-----	—+	Yes	Yes
ERAP1	++++++	--++	First in 2022	Yes
ERAP2	-----	—	First in 2022	Yes
GPLD1	+++++	+---+	MAB only	No
DYNLT1	-----	++++	No	No
T _h infiltrates	++++++	NA	N/A	Yes
PSMD11	--+---	---+	Yes	No
CTSW	-+---+	—	No	Yes
FAM216A	++-+++	-+-	No	No
LYZ	+--+++	++++	No	Yes
C3AR1	--+---	--++	Yes	Yes
DCTN5	++++++	—	No	No
DBNDD1	+----+	+--+	No	No
FPR1	++++++	—+	Yes	Yes

CTSS was deemed the top gene that met these criteria. In addition, it is highly expressed in immune cells, especially monocyte lineage cells. CTSS eQTLs were also associated with cancer risk in the UK Biobank and CTSS upregulation/activation is associated with worse outcomes in B cell follicular lymphoma where it was suggested to alter the landscape of MHC class II presented peptides. We have sought to convey this in the revised manuscript as follows:

“However, some TIME eQTLs were associated with both higher expression of the associated gene and worse ICB response, suggesting that these genes could potentially be inhibited to improve anti-tumor immunity. Of the genes meeting these criteria (**Supplementary Table 14**), only CTSS, TREX1 and PSMD11 had small molecule inhibitors available. ”

“CTSS featured prominently in our cancer risk analysis and, unlike TREX1, had not been implicated as a likely target for solid tumor immunotherapy. Furthermore, we observed increased M1 macrophage infiltration in individuals with the CTSS variant in Hugo et al. suggesting that CTSS activity might contribute to remodeling of the TIME (**Figure S6H**). These considerations led us to choose CTSS as our top gene to validate *in vivo*.”

Analysis inconsistency

11) The authors performed many analyses using data from many sources. I appreciate these efforts. In the meantime, some of statistical analyses seem to be inconsistent, and various ad hoc decisions were made. For example, results using a wide range of FDR cutoffs were reported for different analyses, ranging from $FDR < 0.05$, 0.2, 0.25 or 0.5. This gives the reader a feeling of fishing, e.g., if nothing was significant at the FDR 0.05 level, a more liberal FDR level was used in order to report “significant” findings. It would be useful if consistent statistical analyses and criteria can be used.

We have now simplified our analysis framework to use common criteria for all analyses. We have updated the language to ensure that individual SNP associations are only considered significant if they pass an $FDR < 0.05$. Supplementary tables include all associations annotated according to FDR to allow readers to explore other thresholds for individual SNPs. All analyses using polygenic scores use the same approach for SNP selection and polygenic score construction and the resulting polygenic scores are validated in independent cohorts.

Reviewer #4, expert in genomics and cancer immunology (Remarks to the Author):

Pagadala et al. present an interesting analysis of germline genetic variants associated with immune phenotypes, and the overlap of these associations with cancer risk and treatment outcomes in immunotherapy. The authors assess TIME-associated germline variants curated from literature, and identify further SNPs in an analysis of the TCGA database. Out of this set of variants, approx. 20% are shown to be cancer-relevant, i.e. associated with cancer risk or ICB outcome. As an example, CTSS inhibition was shown to improve survival in a mouse model.

The paper is timely and relevant for the field, also because the application of germline genetics is still difficult in the field of cancer immunotherapies, where large cohorts for hypothesis-free analyses (e.g. GWAS) are not yet available / of sufficient size. Thus, starting with an immune-relevant subset of variants is a promising approach.

We thank the reviewer for the positive assessment and helpful feedback.

That said, there are a couple of concerns I'd like to address:

Major:

- The figures and overall presentation of the paper suggest that "all associations are created equal" (e.g. Figure 3A), although different FDR cutoffs for multiple testing correction were used, depending on the analysis and (likely) power considerations. I don't think that is entirely appropriate. While I wouldn't necessarily insist on Bonferroni corrections in a discovery setting where I'm interested in overlapping associations for multiple phenotypes, some choices appear overly lenient (FDR-corrected $p=0.25$ or even $p=0.5$ for ICB outcomes).

In the revised manuscript we have ensured that all statistical cutoffs are consistent. We have switched to a shrinkage-based polygenic score construction approach to avoid using arbitrary FDR cutoffs to nominate eQTLs for inclusion.

Minor:

- The authors use the term "IP component" to describe the set of variables they test for association with germline genetic variation. In the majority of cases, these components are just RNA expression levels of single genes. Wouldn't it be more straightforward to just refer to eQTL analyses instead, and to use the term IP components only for the more complex variables? For example, on page 7 the authors refer to "GWAS analysis for 17 IP components corresponding to genes in the HLA region of chromosome 6" -> "GWAS for HLA gene expression levels" or similar wouldn't require reading it several times.

We now refer to associations affecting gene expression levels as eQTLs throughout the manuscript.

- Figure 1A is very helpful to better understand the analysis strategy, but it is sometimes difficult to follow the logic in the text. For example, on page 7 the authors describe 825 associations with 75 IP components, but then write that those are based on a suggestive multiple testing threshold, with only a subset reaching genome-wide significance. Given that there is no scarcity of significant associations, it might be better to start with the genome-wide significant associations, and then expand.

We thank the review for this suggestion. Our revised manuscript now uses polygenic scores that perform SNP selection. As such, we felt it was simpler to move forward with the more permissive set of candidate TIME eQTLs and allow the polygenic score and cancer phenotype association studies to focus our attention on the most interesting eQTLs.

- HLA genes eQTL analysis: HLA-DRB3,4,5 were included, but they are copy-number variable and in LD with specific HLA-DRB1 alleles. Did the authors check if expression estimates are credible (e.g.: there should be a significant number of individuals with no HLA-DRB3,4, or 5 expression).

Figure R8 Boxplot of HLA-DRB1 and HLA-DRB5 in TCGA.

We thank the reviewer for highlighting this concern. HLA-DRB3 and HLA-DRB4 were not present in the Pancan Atlas RNA-seq dataset and thus were not analyzed in our study. We revisited the expression data to confirm that levels of HLA-DRB5 were significantly lower than HLA-DRB1 as expected. The Pancan Atlas RNA-seq was harmonized to ensure proper alignment and removal of batch effects. In addition, associations with HLA-DRB5 were reported in other germline variant studies of the TCGA (Sayaman et al) and other large GWAS studies, thus we included it in our revised analysis (Su et al.)^{1,19}.

- Figure 4C: p values missing

We have ensured that all p-values are reported in Figure 4.

- Figure 5E: Are the p values for the pairwise comparisons switched? (low/medium and medium/high)

We have taken care to correct any p-value switches.

- I appreciate the additional functional analysis component of the paper, but the CTSS analyses appear a bit out of place. It's true that there's not much known about CTSS in solid tumors, but the results are also not surprising given what was shown previously e.g. for follicular lymphoma. Also, CTSS has been investigated in autoimmune settings, showing consequences on macrophage function and gene expression levels (Review by Brown et al., Respiratory Research, 2020). In solid tumors, CTSS inhibition was shown to impact TGF-beta-related pathways (Li et al., J Cancer, 2021), autophagy (Fei et al., Front Oncol, 2020) and BRCA1 stability (Kim et al., Cell Death and Differentiation, 2019). So there is a lot to consider when investigating CTSS on a functional level in the context of immune-checkpoint blockade. That said, the results related to combination therapy with anti-PD1 are interesting, and I'm aware that reviewers tend to ask for additional "functional insight" in manuscripts focused on statistical associations.

We thank the reviewer for highlighting the additional context surrounding CTSS and have incorporated these references into our discussion as follows:

“In solid tumors, reports have highlighted that CTSS can impact TGF β -related activities²⁰, autophagy²¹ and BRCA1 stability²², so it is possible that the effects of inhibiting CTSS are not exclusive to the tumor immune microenvironment. Nonetheless, we observed remodeling of the suppressive and inflammatory-like macrophage populations in the mouse tumors treated with CTSS inhibitor.”

- About a third of the introduction is already a summary of the results. Maybe this part can be shortened.

We shortened the summary of the results in the introduction.

- Abstract: Too many numbers and a bit confusing. E.g.: "we evaluated 194 literature-curated TIME associations and 890 associations detected with 157 immune phenotype (IP) components". Maybe just write "1084 associations of TIME SNPs with..." and explain later? Also, "IP components" should be clearly defined if used in the abstract.

We have revised the abstract, with attention to conveying only essential details.

References

1. Germline genetic contribution to the immune landscape of cancer. *Immunity* **54**, 367–386.e8 (2021).
2. Jafarpour, S. *et al.* Association of rare variants in genes of immune regulation with pediatric autoimmune CNS diseases. *J. Neurol.* (2022) doi:10.1007/s00415-022-11325-2.
3. Liu, P. *et al.* Rare Variants in Inborn Errors of Immunity Genes Associated With Covid-19 Severity. *Front. Cell. Infect. Microbiol.* **12**, 888582 (2022).
4. Grulich, A. E., van Leeuwen, M. T., Falster, M. O. & Vajdic, C. M. Incidence of cancers in people with HIV/AIDS compared with immunosuppressed transplant recipients: a meta-analysis. *Lancet* **370**, 59–67 (2007).
5. Yang, J., Lee, S. H., Goddard, M. E. & Visscher, P. M. GCTA: a tool for genome-wide complex trait analysis. *Am. J. Hum. Genet.* **88**, 76–82 (2011).
6. Yang, J. *et al.* Common SNPs explain a large proportion of the heritability for human height. *Nat. Genet.* **42**, 565–569 (2010).
7. Seabold, S. & Perktold, J. Statsmodels: Econometric and Statistical Modeling with Python. *Proceedings of the 9th Python in Science Conference* Preprint at <https://doi.org/10.25080/majora-92bf1922-011> (2010).
8. Thorsson, V. *et al.* The Immune Landscape of Cancer. *Immunity* **51**, 411–412 (2019).
9. Danaher, P. *et al.* Gene expression markers of Tumor Infiltrating Leukocytes. *J Immunother Cancer* **5**, 18 (2017).
10. Newman, A. M. *et al.* Determining cell type abundance and expression from bulk tissues with digital cytometry. *Nat. Biotechnol.* **37**, 773–782 (2019).
11. Knight, J. *et al.* Conditional analysis identifies three novel major histocompatibility complex loci associated with psoriasis. *Hum. Mol. Genet.* **21**, 5185–5192 (2012).
12. Tian, C. *et al.* Genome-wide association and HLA region fine-mapping studies identify susceptibility loci for multiple common infections. *Nat. Commun.* **8**, 599 (2017).
13. Aguiar, V. R. C., Masotti, C., Camargo, A. A. & Meyer, D. HLApers: HLA Typing and Quantification of Expression with Personalized Index. *Methods Mol. Biol.* **2120**, 101–112 (2020).
14. Martín-Cófreces, N. B. & Sánchez-Madrid, F. Sailing to and Docking at the Immune Synapse: Role of

- Tubulin Dynamics and Molecular Motors. *Front. Immunol.* **9**, 1174 (2018).
15. Paladini, F. *et al.* An allelic variant in the intergenic region between ERAP1 and ERAP2 correlates with an inverse expression of the two genes. *Sci. Rep.* **8**, 10398 (2018).
 16. Speed, D., Hemani, G., Johnson, M. R. & Balding, D. J. Improved heritability estimation from genome-wide SNPs. *Am. J. Hum. Genet.* **91**, 1011–1021 (2012).
 17. Visscher, P. M. *et al.* Statistical power to detect genetic (co)variance of complex traits using SNP data in unrelated samples. *PLoS Genet.* **10**, e1004269 (2014).
 18. Liu, J. *et al.* An Integrated TCGA Pan-Cancer Clinical Data Resource to Drive High-Quality Survival Outcome Analytics. *Cell* **173**, 400–416.e11 (2018).
 19. Su, W.-M. *et al.* Association Analysis of , and Polymorphisms in Chinese Patients With Parkinson's Disease and Multiple System Atrophy. *Front. Genet.* **12**, 765833 (2021).
 20. Wei, L., Shao, N., Peng, Y. & Zhou, P. Inhibition of Cathepsin S Restores TGF- β -induced Epithelial-to-mesenchymal Transition and Tight Junction Turnover in Glioblastoma Cells. *J. Cancer* **12**, 1592–1603 (2021).
 21. Fei, M. *et al.* Inhibition of Cathepsin S Induces Mitochondrial Apoptosis in Glioblastoma Cell Lines Through Mitochondrial Stress and Autophagosome Accumulation. *Front. Oncol.* **10**, 516746 (2020).
 22. Kim, E. H., Wong, S.-W. & Martinez, J. Programmed Necrosis and Disease: We interrupt your regular programming to bring you necroinflammation. *Cell Death Differ.* **26**, 25–40 (2019).

Reviewers' Comments:

Reviewer #1:

Remarks to the Author:

The authors have addressed all my concerns. I suggest acceptance.

Reviewer #3:

Remarks to the Author:

The authors made good efforts on addressing the issues raised in the previous review. The revision is improved. There are a few issues that remain to be addressed.

Major:

- In the abstract, "Specifically, we achieved an AUC of at least 0.7 in predicting immunotherapy response using only germline genetic variants." Such AUC results were provided in the main text and hence this statement is not supported.
- "No composite phenotypes passed heritability thresholds and thus remaining associations were with gene expression and will be referred to as TIME eQTLs." Can you comment on why the composite phenotypes didn't pass the heritability thresholds? Can you evaluate whether heritability varies by tumor types?
- The authors showed that individuals in the top decile PRS of cancer risk/survival and ICB have better outcomes compared to those in the lowest decile. What about the performance of intermediate PRS values? The author should at least show the complete association across all PRS deciles in the supplementary materials.
- It is not clear how the AUC in the ROC analyses of PRS on ICB response was calculated. Was it based on PRS + age + gender + 10 PCs or only PRS? Was it calculated using all samples or only 30% of the samples that were held off? It would be useful if the authors can provide the incremental value of adding PRS to key baseline covariates and compare the AUC to those calculated using clinicopathological features. This will help show that PRS of ICB responses has additional clinical values.
- In the methods section, the author mentions "METAL analysis of response (iRecist: CR, PR, SD)" . Did you exclude progression disease (PD)? Please spell out what CR, PR and SD stand for. In addition, the definition of response should be made clear. If the objective response rate was used as a binary outcome and modeled using logistic regression, it should be clearly stated in the methods section.

Minor:

- Figure 4C-D, it may be better to change the unit of x-axis to months
- The authors should make it clear how many of the SNPs were included in each PRS calculation.
- It will be helpful if the authors could incorporate samples size used in each analysis in Figure 1.

Reviewer #4:

Remarks to the Author:

The revised version of the manuscript has improved a lot and reads well, and I'd like to thank the

authors for the detailed responses to the reviewers' comments.

Cleaning up the methods and descriptions, as well as using consistent statistical cutoffs resulted in a manuscript that is much easier to comprehend and evaluate.

I only have one minor concern, related to the HLA-DRB5 eQTL analysis. It is expected that expression level is lower when compared to HLA-DRB1, but it is also expected that a significant number of patients should have no HLA-DRB5 expression at all, since the gene is copy number variable (0,1,2) and is only present on haplotypes together with HLA-DRB1*15 and HLA-DRB1*16 alleles. Homology with HLA-DRB3 and HLA-DRB4 (both not present in the reference dataset and in strong LD with other HLA-DRB1 alleles) makes it difficult to trust the read alignments. This should be flagged as a potential bias. Alternatively, the authors could only include patients with HLA-DRB1*15 and HLA-DRB1*16 allele calls in this analysis, to ensure that reads were mapped to a gene that is actually present in the respective patients' genomes

We would like to thank all reviewers for their helpful feedback throughout the review process. We are very happy with the improvements to the manuscript that resulted from their excellent suggestions.

When validating our updated code, we found and corrected two minor errors that affected Figures 4A-B and the small right panel in 5E respectively.

Figure 4A-B: Correcting the code resulted in a small shift in effect sizes leading to a change in the order of tumors along the y-axis and the PSS score reaching statistical significance in additional tumor types (LUSC, ESCA, READ and PAAD) in Figure 4A after multiple testing correction (red versus blue color) and losing significance in SKCM. This change does not affect the overall interpretation of the results and we have corrected the figure and associated text listing significant tumor types in this revised version of the manuscript.

Figure 4A and B compared between 1st and 2nd revision:

Figure 5E: In the previous version, the small panel for the clinical variables on the right side plotted the incorrect size for the underlying data. This has been corrected, and to ensure that interpretation is not solely dependent on the plot which gives a more qualitative representation, we included the effect sizes and p-values associated with Figure 5E as Table S15. This did not affect the manuscript text.

Fig 5E right side at Rev 1

The code for both figures has been corrected in the Github repository.

Reviewer #1 (Remarks to the Author):

The authors have addressed all my concerns. I suggest acceptance.

Reviewer #3 (Remarks to the Author):

The authors made good efforts on addressing the issues raised in the previous review. The revision is improved. There are a few issues that remain to be addressed.

Major:

- In the abstract, “Specifically, we achieved an AUC of at least 0.7 in predicting immunotherapy response using only germline genetic variants.” Such AUC results were provided in the main text and hence this statement is not supported.

We have revised our abstract as follows:

“Polygenic immunotherapy response scores achieved an ROC AUC greater than 0.7 in two independent validation cohorts.”

We have ensured that the ROC AUC is clearly reported in the manuscript. It is shown in Figure 5C and described in the manuscript as follows:

“We validated the predictive potential for this polygenic score in two independent cohorts, one consisting of renal cell carcinomas, and the other of non-small cell lung cancers. In both cohorts, responders had significantly higher polygenic ICB scores (PICS) (Figure 5A-B) and in ROC analysis the PICS achieved an area under the curve greater than 0.7 (Figure 5C).”

- “No composite phenotypes passed heritability thresholds and thus remaining associations were with gene expression and will be referred to as TIME eQTLs.” Can you comment on why the composite phenotypes didn’t pass the heritability thresholds? Can you evaluate whether heritability varies by tumor types?

We thank the reviewer for this suggestion. We believe that associations with composite phenotypes are more challenging to detect for two reasons. The first is that the composite value

is likely noisier as it represents a proxy measurement derived from multiple gene expression values. The second is that by going from single gene eQTL to essentially a multi-gene eQTL we are increasing the polymorphicity (i.e. the number of SNPs influencing the trait). This likely reduces the effect size of individual SNPs relative to the phenotype variable¹. Consequently, detecting these associations may require much larger sample sizes, and possibly more accurate measurements of e.g. immune cell infiltration levels.

We have added this to the discussion:

“Phenotypes comprising multiple genes are likely to have higher polymorphicity, which could make detection of associations with composite phenotypes such as expression-based estimates of pathway activity or immune cell infiltrates more difficult.”

Guidance for heritability estimation is to use at least 3160 unrelated samples in order to get the standard error down to 0.1². Most tumor types in the TCGA dataset have significantly fewer samples. We sought to evaluate heritability solely within breast cancer, which has the most samples, however we note that the different subtypes of breast cancer tend to have very different levels of immune infiltration³.

We replicated heritability analysis pipeline with only TCGA BRCA patients controlling for age, sex and ER/PR/HER2 receptor status. After 2-state GCTA analysis, 17 genes were heritable (FDR < 0.05), less than the pancancer analysis likely due to reduced sample sizes. The majority of these genes were MHC Class I and II genes (HLA-A, HLA-C, HLA-G, HLA-DRB1, HLA-DRB5, HLA-DQB1, HLA-DQB2, MR1, MICA, BTN3A2, HLA-DQA2, HLA-DQA1, PAICS), consistent with the pancancer heritability analysis. *ERAP2* and *DCTN5* were both heritable in BRCA and pancancer analysis. *KRR1* and *FN1* were heritable only in BRCA analysis. *FN1* encodes fibronectin, which could play a key role in stromal microenvironment and tumor invasion⁴. It has been implicated in development of several tumors, including breast cancer^{5,6}. *KRR1* is a proteasomal subunit that might be related to integrin expression in breast cancer⁷. These results suggest that there are likely shared heritable features related to antigen presentation, but also differences that could be unique to each cancer’s microenvironment.

We have added the following to the manuscript:

*“To assess the possibility of tumor-type specific SNP-heritable effects, we revisited the SNP-heritability analysis in breast cancer, which had the most samples. The 2-state heritability analysis uncovered 17 genes (FDR < 0.05), including HLA region genes (HLA-A, HLA-C, HLA-G, HLA-DRB1, HLA-DRB5, HLA-DQB1, HLA-DQB2, MR1, MICA, BTN3A2, HLA-DQA2, HLA-DQA1, PAICS) and ERAP2 and DCTN5 genes which were shared with the pancancer analysis. Two additional genes, KRR1 and FN1, were only detected in the breast cancer-specific analysis. FN1 encodes fibronectin, which plays a role in the stromal microenvironment and tumor invasion³¹. It has been implicated in development of several tumors, including breast cancer^{32,33}. KRR1 is a proteasomal subunit linked to integrin expression in breast cancer³⁴. These results suggest that there are likely shared heritable features related to antigen presentation, but also differences that could be unique to each cancer’s microenvironment (**Supplementary Table 4**). However, larger sample sizes are needed to investigate tumor-type specific effects.”*

And in methods:

“We repeated the 2-state analysis for breast cancer only samples using age, ER, PR and HER2+ status as a covariate. Hormone receptor status was retrieved from clinical files describing IHC results.”

- The authors showed that individuals in the top decile PRS of cancer risk/survival and ICB have better outcomes compared to those in the lowest decile. What about the performance of intermediate PRS values? The author should at least show the complete association across all PRS deciles in the supplementary materials.

We thank the reviewer for this suggestion. We have now added plots showing all deciles for melanoma and prostate cancer PRS in the validation cohorts to the supplementary materials. The plots are also shown here.

Figure S5: TIME eQTLs underlying antigen presentation stratify melanoma and prostate cancer risk. (A) Odds of melanoma risk among individuals in PRS (# snps =43) quantiles in High Density Melanoma cohort. (B) Odds of prostate cancer risk among individuals in the top and bottom 10th quantile of PRS (# snps=26) in ELLIPSE consortium. Error bars indicate 95% confidence interval.

We have added the full range of PRS deciles as Figure S5A and S5B.

- It is not clear how the AUC in the ROC analyses of PRS on ICB response was calculated. Was it based on PRS + age + gender + 10 PCs or only PRS? Was it calculated using all samples or only 30% of the samples that were held off? It would be useful if the authors can provide the incremental value of adding PRS to key baseline covariates and compare the AUC to those calculated using clinicopathological features. This will help show that PRS of ICB responses has additional clinical values.

We have clarified in the methods and figure caption that the ROC curve in Figure 5C is based solely on the PRS score. Based on this reviewer’s comment, we sought to evaluate how this score compared to scores based solely on clinical covariates or established biomarkers such as TMB and checkpoint gene expression, and whether combining these sources of information was beneficial. The only clinical covariates that are available for all ICB treated cohorts were sex and age, which did not effectively predict ICB response. Combining age and sex with PRS scores did not improve over PRS scores alone (now shown in Figure S7 panels C and D).

Figure S7: TIME eQTLs implicate targets for modulating immune responses, Related to Figure 5. (C) ROC-AUC analysis of PICS, clinical variables (age, sex) and PICS+clinical variables in Rizvi *et al.* validation cohorts. **(D)** ROC-AUC analysis of PICS, clinical variables (age, sex) and PICS+clinical variables in Rizvi *et al.* validation cohorts.

We also evaluated PRS relative to TMB and checkpoint gene expression in the Miao cohort, one of the independent validation cohorts where we did not fit model parameters. In this cohort, the PICS performed similarly to the combination of TMB and checkpoint gene expression for informing ICB response, and adding germline PRS to a baseline model (TMB/PD-1/PD-L1/CTLA4) resulted in a model that better explained ICB outcomes (i.e. significantly improved the variance explained; anova $p < 0.007$).

Figure S7: TIME eQTLs implicate targets for modulating immune responses, Related to Figure 5. (F) Variance explained (pseudo-R²) by TMB, TMB/PD1/PDL1/CTLA4, PICS and PICS/TMB/PD1/PDL1/CTLA4 in Miao *et al.*

We have updated the manuscript to describe these analysis as follows:

“Polygenic ICB scores (PICS) were constructed from nominally significant TIME eQTLs identified in the METAL analysis of response (iRecist: CR, PR, SD) across four ICB-treated melanoma cohorts (Van Allen, Hugo, Riaz, and Snyder). The PICS model (number of SNPs=31) was validated on two independent ICB treated cohorts (Rizvi and Miao). ROC-AUC and Mann-Whitney U tests¹⁵⁴

were the primary evaluation metrics used to assess PICS performance for predicting ICB response. We further conducted ROC-AUC analysis with clinical variables (Age, Sex) alone, PICS alone, and PICS with clinical variables. Logistic regression was used to estimate the variance in response status explained by PICS, TMB and checkpoint gene expression. McFadden pseudo-R2 was reported and models were compared by anova.”

- In the methods section, the author mentions” METAL analysis of response (iRecist: CR, PR, SD)” . Did you exclude progression disease (PD)? Please spell out what CR, PR and SD stand for. In addition, the definition of response should be made clear. If the objective response rate was used as a binary outcome and modeled using logistic regression, it should be clearly stated in the methods section.

We have now clarified classification for responder status in Methods (Immunotherapy Response Analysis):

“Response phenotypes were determined from iRECIST criteria¹⁶⁹. Patients were categorized as responders if they had iRECIST criteria: CR (complete response), PR (partial response) and SD (stable disease). Non-responders had iRECIST criteria: PD (progressive disease). This resulted in 114 responders and 165 non-responders.”

Minor:

- Figure 4C-D, it may be better to change the unit of x-axis to months

We have updated Figure 4C-D to use months.

- The authors should make it clear how many of the SNPs were included in each PRS calculation.

We updated the methods and added the number of SNPs included in each model in the main figures:

“The melanoma risk model (number of SNPs=43) was validated using the Geneva melanoma cohort (excluding individuals with no FH of melanoma), while the prostate cancer risk model (number of SNPs=26) was validated on all individuals in the ELLIPSE prostate cancer cohort.”

“The PSS model for TCGA LUAD (number of SNPs=28) was validated in the Sherlock cohort.”

“The PICS model (number of SNPs=32) was validated on two independent ICB treated cohorts (Rizvi and Miao).”

Updated PRS Figure

Updated PSS Figure:

Updated PICS Figure

- It will be helpful if the authors could incorporate samples size used in each analysis in Figure 1.

We have now updated Figure 1 with sample sizes for each analysis.

Sample sizes in the TCGA survival analysis varied by tumor type which we now report in Methods. To avoid crowding the figure, we did not include these in Figure 1. We also included the number of samples withheld to validate the PSS score for each TCGA tumor type in Figure 4A and B.

Reviewer #4 (Remarks to the Author):

The revised version of the manuscript has improved a lot and reads well, and I'd like to thank the authors for the detailed responses to the reviewers' comments.

Cleaning up the methods and descriptions, as well as using consistent statistical cutoffs resulted in a manuscript that is much easier to comprehend and evaluate.

I only have one minor concern, related to the HLA-DRB5 eQTL analysis. It is expected that expression level is lower when compared to HLA-DRB1, but it is also expected that a significant number of patients should have no HLA-DRB5 expression at all, since the gene is copy number variable (0,1,2) and is only present on haplotypes together with HLA-DRB1*15 and HLA-DRB1*16 alleles. Homology with HLA-DRB3 and HLA-DRB4 (both not present in the reference dataset and in strong LD with other HLA-DRB1 alleles) makes it difficult to trust the read alignments. This should be flagged as a potential bias. Alternatively, the authors could only include patients with HLA-DRB1*15 and HLA-DRB1*16 allele calls in this analysis, to ensure that reads were mapped to a gene that is actually present in the respective patients' genomes.

We have now further investigated HLA-DRB5 associations. As suggested by the reviewer we flagged the original HLA-DRB5 association as potentially biased in our supplemental table and then repeated the conditional GWAS analysis with HLA-DRB5 expression using only individuals with HLA-DRB1*15 and HLA-DRB1*16 (n=1564) to avoid confounding effects of HLA-DRB3&4 homology. We identified 2 variants that remained statistically significant and included them in Supplementary Table 6 with details. Revisiting our analysis, the 2 HLA-DRB5 variants were not associated with risk or survival. These SNPs did not need the MAF inclusion criteria for the ICB association analysis. We have updated the manuscript as follows:

*“We note that HLA-DRB5 only occurs on specific haplotypes, but has homology to HLA-DRB3 and HLA-DRB4 which could lead to erroneous assignment of gene expression in individuals where the HLA-DRB5 gene is absent. We therefore revisited eQTL analysis for HLA-DRB5 using only individuals with HLA-DRB1*15 and HLA-DRB1*16 alleles, which indicate haplotypes inclusive of the HLA-DRB5 gene^{41–43}. This analysis implicated 2 SNPs associated with HLA-DRB5 expression levels. (**Supplementary Table 6**).”*

We also updated methods:

*“Analysis for HLA-DRB5 was revisited using only individuals with HLA-DRB1*15 and HLA-DRB1*16 allele calls indicating haplotypes where the HLA-DRB5 gene is present. We re-ran conditional GWAS analysis only within individuals with these alleles (n=1564). SNPs with Bonferroni-corrected p-value ($p < 1 \times 10^{-5}/17$) were kept for further analysis.”*

Works Cited:

1. Wu, T., Liu, Z., Mak, T. S. H. & Sham, P. C. Polygenic power calculator: Statistical power and polygenic prediction accuracy of genome-wide association studies of complex traits. *Front. Genet.* **13**, 989639 (2022).
2. Visscher, P. M. *et al.* Statistical power to detect genetic (co)variance of complex traits using SNP data in unrelated samples. *PLoS Genet.* **10**, e1004269 (2014).
3. Dieci, M. V., Miglietta, F. & Guarneri, V. Immune Infiltrates in Breast Cancer: Recent Updates and Clinical Implications. *Cells* **10**, (2021).
4. Wang, Y., Xu, H., Zhu, B., Qiu, Z. & Lin, Z. Systematic identification of the key candidate genes in breast cancer stroma. *Cell. Mol. Biol. Lett.* **23**, 44 (2018).
5. Korah, R., Boots, M. & Wieder, R. Integrin $\alpha 5\beta 1$ Promotes Survival of Growth-Arrested Breast Cancer Cells. *Cancer Research* vol. 64 4514–4522 Preprint at <https://doi.org/10.1158/0008-5472.can-03-3853> (2004).
6. Sun, Y. *et al.* High expression of fibronectin 1 indicates poor prognosis in gastric cancer. *Oncol. Lett.* **19**, 93–102 (2020).
7. Lu, S., Simin, K., Khan, A. & Mercurio, A. M. Analysis of Integrin $\beta 4$ Expression in Human Breast Cancer: Association with Basal-like Tumors and Prognostic Significance. *Clinical Cancer Research* vol. 14 1050–1058 Preprint at <https://doi.org/10.1158/1078-0432.ccr-07-4116> (2008).
8. Korah, R., Boots, M. & Wieder, R. Integrin $\alpha 5\beta 1$ promotes survival of growth-arrested breast cancer cells: an in vitro paradigm for breast cancer dormancy in bone marrow. *Cancer Res.* **64**, 4514–4522 (2004).
9. Mann, H. B. & Whitney, D. R. On a Test of Whether one of Two Random Variables is Stochastically Larger than the Other. *The Annals of Mathematical Statistics* vol. 18 50–60 Preprint at <https://doi.org/10.1214/aoms/1177730491> (1947).
10. Seymour, L. *et al.* iRECIST: guidelines for response criteria for use in trials testing immunotherapeutics. *Lancet Oncol.* **18**, e143–e152 (2017).
11. Miretti, M. M. *et al.* A High-Resolution Linkage-Disequilibrium Map of the Human Major Histocompatibility Complex and First Generation of Tag Single-Nucleotide Polymorphisms. *Am. J. Hum. Genet.* **76**, 634 (2005).
12. Furukawa, H. *et al.* The role of common protective alleles HLA-DRB1*13 among systemic autoimmune diseases. *Genes Immun.* **18**, 1–7 (2017).
13. Degenhardt, F. *et al.* Construction and benchmarking of a multi-ethnic reference panel for the imputation of HLA class I and II alleles. *Human Molecular Genetics* vol. 28 2078–2092 Preprint at <https://doi.org/10.1093/hmg/ddy443> (2019).

Reviewers' Comments:

Reviewer #3:

None

Reviewer #4:

Remarks to the Author:

The authors have addressed my concerns. I suggest acceptance.